# *MicroRNA-203* represses selection and expansion of oncogenic *Hras* transformed tumor initiating cells

**Kent Riemondy[1], Xiao-jing Wang[2], Enrique C Torchia[3,4], Dennis R Roop[3,4], Rui Yi[1]\***

[1]Department of Molecular, Cellular and Developmental Biology, University of Colorado, Boulder, Boulder, United States; [2]Department of Pathology, University of Colorado Denver Anschutz Medical Campus, Denver, United States; [3]Department of Dermatology, University of Colorado Denver Anschutz Medical Campus, Denver, United States; [4]Charles C Gates Center for Regenerative Medicine and Stem Cell Biology, University of Colorado Denver Anschutz Medical Campus, Denver, United States

**Abstract** In many mouse models of skin cancer, only a few tumors typically form even though many cells competent for tumorigenesis receive the same oncogenic stimuli. These observations suggest an active selection process for tumor-initiating cells. Here, we use quantitative mRNA- and miR-Seq to determine the impact of $Hras^{G12V}$ on the transcriptome of keratinocytes. We discover that *microRNA-203* is downregulated by $Hras^{G12V}$. Using a knockout mouse model, we demonstrate that loss of *microRNA-203* promotes selection and expansion of tumor-initiating cells. Conversely, restoration of *microRNA-203* using an inducible model potently inhibits proliferation of these cells. We comprehensively identify *microRNA-203* targets required for *Hras*-initiated tumorigenesis. These targets include critical regulators of the *Ras* pathway and essential genes required for cell division. This study establishes a role for the loss of *microRNA-203* in promoting selection and expansion of *Hras* mutated cells and identifies a mechanism through which *microRNA-203* antagonizes *Hras*-mediated tumorigenesis.

\*For correspondence: yir@colorado.edu

**Competing interests:** The authors declare that no competing interests exist.

## Introduction

Recent efforts in comprehensively sequencing human cancer genomes have confirmed ~140 protein-coding genes that, when mutated, can drive tumorigenesis (*Vogelstein et al., 2013*). When genome sequencing data were utilized to construct the history of cancer cells in breast cancer, it was revealed that a considerable amount of 'molecular time' exists between the common ancestors that harbor the great majority of driver mutations and the phenotypically identified cancer cells that compose the bulk of the tumor (*Nik-Zainal et al., 2012*). In support of these observations, lineage tracing experiments conducted in genetically engineered mouse models revealed that only a few clones give rise to tumors whereas a vast majority of mutated cells are unable to sustain tumorigenesis (*Driessens et al., 2012*; *Schepers et al., 2012*). These results suggest that even after the acquisition of key driver mutations in the nascent cancer cells, these cells must still undergo continuous evolution and likely clonal selection before developing into clinically apparent tumors. To begin to understand the molecular basis underlying such selection, we examined papilloma formation driven by oncogenic *Hras* in the skin, a well-characterized model where *Hras* has been shown to initiate the formation of tumors that clonally evolve (*Brown et al., 1986*; *Driessens et al., 2012*; *Beck and Blanpain, 2013*). Oncogenic *Ras* mutations are some of the most frequently detected driver mutations in human cancer. Among the three *Ras* genes (*H-*, *K-*, and *N-ras*), *Hras* is commonly mutated in tumors originated from stratified

**eLife digest** DNA mutations occur and accumulate during an individual's lifetime. Often these changes are harmless. But some mutations—called driver mutations—can trigger the formation of tumors. This is often because these mutations allow the cells to grow faster than normal cells. Mutations in genes in the *Ras* gene family are among the most common driver mutations found in human cancers. These common mutations lead to the uncontrolled activation of genes that are normally tightly controlled, which in turn allows the cells to divide more and live for longer: these are two key features of cancer cells.

So, how are *Ras* genes and the genes that they control regulated to prevent such dangerous over activation? One mechanism rests on binding sites in their messenger RNA sequence that are recognized by smaller RNA molecules called microRNAs. RNA molecules are created when genes are transcribed. Some RNAs, called messenger RNAs, are then decoded to create proteins. Many other RNAs, including microRNAs, do not code for proteins, but instead bind to many messenger RNA targets, and repress their ability to be decoded into proteins. Three genes, called *Hras*, *Kras*, and *Nras*, are regulated in this way by numerous microRNAs, which together act to dampen the normal activities of these genes.

Riemondy et al. investigate how a cancer-promoting mutation in the *Hras* gene affects the activities of microRNAs in mouse skin cells in culture. By measuring RNA levels, the experiments reveal that skin cells carrying this mutation produce significantly lower levels of what is normally the most highly produced microRNA in the skin. This microRNA, called *microRNA-203*, acts to limit the proliferation of skin cells when these cells are dividing rapidly. When the gene encoding *microRNA-203* was deleted in mice, the skin cells proliferated more. These mice also developed more skin tumors than normal mice when they were exposed to cancer-causing chemicals. When the gene for *microRNA-203* was added into skin cells carrying the *Hras* mutation and then activated, the cells both divided less and, as a results, grew less. This indicates that *microRNA-203* could prevent cancerous cells from expanding in number, a key event in the initiation of tumors.

Riemondy et al. also used a variety of approaches to identify the molecules targeted by *microRNA-203* in the skin, and reveal that it targets multiple signaling pathways, including components of the Ras pathway, to suppress cell proliferation. Together, these findings highlight *microRNA-203* as a potential source of new treatments to prevent or slow tumor growth in humans.

epithelial tissues including squamous cell carcinoma in the skin, head, and neck cancer as well as bladder cancer (*Bos, 1989*; *Agrawal et al., 2011*; *Stransky et al., 2011*). Experimental and genomic sequencing studies have revealed that the vast majority of *Ras* mutations are missense, point mutations at amino acid residues glycine 12 (G12), glycine 13 (G13), or glutamine 61 (Q61) (*Bos, 1989*). Structural and biochemical studies have further confirmed that all of these mutations generally interfere with the GTP binding pocket and compromise the GTPase activity of *Ras* proteins. In turn, these mutations lead to uncontrolled activation of downstream effectors including Raf/MEK/ERK and PI(3)K pathways, resulting in sustained cell survival and proliferation observed in human cancers. Because of the prominent role of *Ras* mutations in human cancer, extensive efforts have been devoted to uncover and subsequently target downstream pathways that are regulated by *Ras* mutations. However, the immediate impact of *Ras* mutations on the transcriptome, in particular, with regards to microRNAs (miRNAs) remains unclear.

miRNAs are a class of small, noncoding RNA species that are involved in virtually all biological processes examined in mammals including mouse and human. These regulatory RNA molecules function by repressing the protein producing ability of mRNA targets through destabilization of mRNAs and inhibition of translation (*Bartel, 2009*). miRNAs typically target a large number of mRNAs in a dosage- and cell context-dependent manner (*Mukherji et al., 2011*). As prominent proto-oncogenes, *Ras* mutations have long been recognized to interact with the miRNA pathway. Indeed, *Hras*, *Kras*, and *Nras* all harbor multiple binding sites for the *let-7* miRNA, a founding member of miRNAs, in their 3′UTRs (*Johnson et al., 2005*). Additionally, impaired miRNA biogenesis in the form of *Dicer1* disruption has been shown to be a tumor-suppressing mechanism for the development of *Kras*-induced lung cancer in a mouse model (*Kumar et al., 2007*). A number of individual miRNAs

were also found to function as modifiers for *Ras*-induced tumorigenesis that include *miR-21, -29,* and *miR-17~92* as tumor-promoting miRNAs and *miR-34, -15/16,* and *miR-143/145* as tumor-suppressing miRNAs (*Kasinski and Slack, 2010*; *Iorio and Croce, 2012*; *Mendell and Olson, 2012*). Collectively, these seminal studies demonstrate unequivocally that the miRNA pathway and individual miRNAs play important roles in *Ras*-induced tumorigenesis. However, it is unclear how *Ras* mutations, usually the tumor-initiating drivers, directly alter the landscape of miRNA expression during tumorigenesis. Importantly, it is also unknown whether the changes in miRNA expression play a role in the selection of oncogenic *Ras*-transformed cells during tumor initiation. Finally, the lack of a comprehensive survey of high-confidence miRNA targets that may play a role downstream of *Ras* mutations hinders our mechanistic understanding and limits the potential to develop miRNA-based therapeutics.

In this study, we utilized our recently improved quantitative miR-Seq techniques to examine the impact of an oncogenic *Hras* mutation (*Hras^{G12V}*) on both mRNA and miRNA expression. We discovered that *miR-203*, the most highly expressed miRNA in the skin (*Yi et al., 2008*; *Jackson et al., 2013*), is downregulated by *Hras^{G12V}*. Using both knockout (KO) and inducible models, we provide evidence for an important role of *miR-203* in restricting expansion of oncogenic *Hras*-transformed cells in vitro and in vivo. We comprehensively surveyed skin-specific targets of *miR-203* and identified a number of novel targets that have important implications for *Hras*-mediated tumorigenesis. Our results suggest that *miR-203* plays a tumor-suppressing role in inhibiting selection and expansion of tumor-initiating cells early in tumor development.

## Results

### *Hras^{G12V}* profoundly deregulates the mRNA and miRNA transcriptome in the skin

Oncogenic mutation of the *Hras* gene is one of the initiating drivers in the development of benign papillomas and malignant squamous cell carcinomas in murine skin chemical carcinogenesis. However, the molecular consequences defining the cellular changes that accompany expansion of oncogenic *Hras*-transformed keratinocytes to initiate papillomas remain elusive. We first investigated the consequences of *Hras^{G12V}* activation on the mRNA transcriptome using a modified form of PolyA+ RNA-Seq, known as 3P-Seq, or 3seq (*Figure 1A–C*). Compared to traditional RNA-Seq, 3Seq allows both quantification of mRNA transcripts and detection of changes in alternative 3′ UTR formation (*Wang et al., 2013*). To examine the immediate impact of *Hras^{G12V}* on primary skin cells, we used primary keratinocytes isolated from newborn skin and performed 3Seq after *Hras^{G12V}* transduction. We did not observe widespread shortening or alternative formation of 3′UTRs, which are often ascribed to oncogenic transformation when comparing tumor cell lines to normal cells (data not shown). This is similar to our previous observation that alternative 3′UTR usage is infrequent within the skin lineages (*Wang et al., 2013*). Over 1100 transcripts were differently expressed (two-fold change and FDR <0.05) in keratinocytes expressing *Hras^{G12V}*, compared to the control (*Figure 1C* and *Figure 1—source data 1*). Gene ontology functional analysis revealed profound deregulation in three core processes by *Hras^{G12V}*: activation of cellular migration, upregulation of pro-angiogenic pathways, and suppression of the terminal differentiation program (*Figure 1D*). All of these three processes are identified as hallmarks of human cancer (*Hanahan and Weinberg, 2011*). The observed widespread changes in the transcriptome also endorse the driver role of *Hras^{G12V}* in skin tumorigenesis. Importantly, transcripts upregulated by *Hras^{G12V}* in our primary keratinocytes strongly and significantly overlapped with the putative cancer stem cell signatures obtained from murine squamous cell carcinoma (SCC) models (*Schober and Fuchs, 2011*). In addition, transcripts upregulated by *Hras^{G12V}* significantly overlapped with transcripts known to be targets of the *c-Fos* transcription factor in a genetic model of SCC (*Durchdewald et al., 2008*). Furthermore, known core components of the *Hras* signaling pathway were also among the differentially detected genes (*Bild et al., 2006*) (*Figure 1E*). These transcriptome data indicate that we have captured the initiating changes induced by oncogenic *Hras* in the keratinocytes.

To define the impact of the oncogenic *Hras* on the landscape of miRNA, we applied our recently developed, quantitative miRNA-Seq (*Zhang et al., 2013*) to *Hras^{G12V}*-transformed keratinocytes. Overall, we detected 15 differentially expressed miRNAs upon *Hras^{G12V}* expression (FDR <0.05, two-fold change) (*Figure 1F–H*). Two key patterns emerged in these profiles. First, the epithelial tissue-specific miRNAs, *miR-203* and *miR-205*, which represent the most abundant miRNAs expressed in

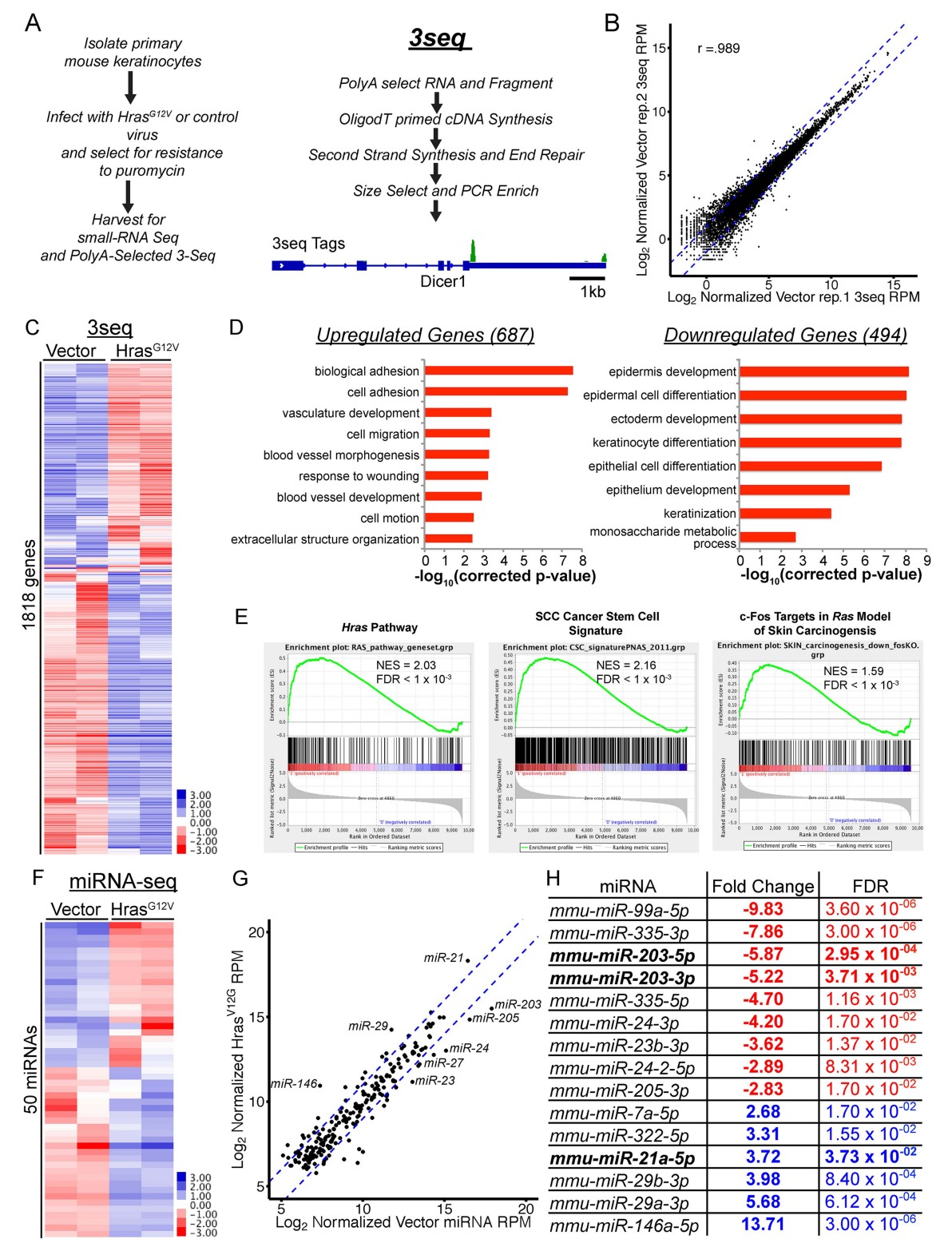

**Figure 1**. Genome-wide profiling of the oncogenic *Hras*<sup>G12V</sup>-transformed miRNA and mRNA transcriptome in primary keratinocytes. (**A**) Schematic of experimental approach to identify deregulated mRNA and miRNA networks driven by oncogenic *Hras*<sup>G12V</sup> using small-RNA Seq and 3Seq. The 3seq library preparation allows quantitative definition of poly-A+ RNA 3'ends and expression levels. (**B**) 3Seq reproducibly detects mRNA expression levels over 4 orders of magnitude. Pearson correlation coefficient displayed (**C**) unsupervised hierarchical clustering of log-transformed mean-centered mRNA

*Figure 1. continued on next page*

Figure 1. Continued

expression levels for all transcripts deregulated twofold by oncogenic $Hras^{G12V}$ (n = 2 libraries per condition) (**D**) Gene Ontology analysis of transcripts up and downregulated by $Hras^{G12V}$ (twofold change FDR <0.05) indicates enrichment for migratory and angiogenic processes, and suppression of keratinocyte differentiation. (**E**) GSEA analysis of selected genesets relevant to skin carcinogenesis. (**F**) Unsupervised hierarchical clustering of log-transformed mean-centered miRNA expression levels for all transcripts deregulated twofold by oncogenic $Hras^{G12V}$ (n = 2 libraries per condition) (**G**, **H**) Abundant miRNAs such as *miR-203*, *miR-205*, and *miR-21* are strongly deregulated by oncogenic *Ras*.

The following source data is available for figure 1:

**Source data 1**. Log2 fold changes for transcripts up or down regulated twofold with FDR <0.05 in $Hras^{G12V}$-transformed keratinocytes.

murine skin and primary keratinocytes were strongly suppressed by $Hras^{G12V}$. Additionally, members of the abundantly expressed, *miR-23/24/27* miRNA cluster were also downregulated by $Hras^{G12V}$. Secondly, *miR-21* was strongly induced becoming the most highly expressed miRNA, consistent with its direct activation by oncogenic *Ras* reported in other systems (*Talotta et al., 2009*). The upregulation of *miR-21* is also consistent with its well-appreciated oncogenic function in skin cancer (*Darido et al., 2011*). *miR-146* was also induced by $Hras^{G12V}$. However, this miRNA is expressed 2-orders of magnitude lower than miR-21, suggesting that its upregulation may have limited contribution to *Hras*-initiated tumorigenesis at this early stage.

We further measured mature *miR-21*, *miR-203*, and *miR-205* RNAs by qPCR. In support of the quantitative performance of our miR-Seq, the differential expression of all three miRNAs measured by qPCR was nearly identical to the quantification by our miR-Seq (*Figure 2A,B*). To initially probe the mechanism through which $HRAS^{G12V}$ suppresses *miR-203* expression, we examined the level of *miR-203* primary transcripts. We previously characterized the transcribed genomic region of *miR-203* including the promoter region and transcription start site (*Jackson et al., 2013*). Because the primary transcript of *miR-203* harbors a polyadenylation [Poly(A)] signal and generates a Poly(A) tail, we directly quantified the abundance of the primary transcripts by counting the 3′end reads of the primary miRNA obtained by 3seq (*Figure 2C*). This result was further confirmed by qPCR measurement specific to the *pri-miR-203* (*Figure 2D*). The degree of downregulation for both mature and primary *miR-203* transcripts was similar as judged by these two independent assays. Collectively, we conclude that the repression of *miR-203* by $Hras^{G12V}$ is most likely mediated by suppressing the production of primary *miR-203* transcripts at an early stage of oncogenic cellular transformation.

## *miR-203* silencing is an early event in mouse and human SCC

Our data revealed the early silencing of *miR-203* by oncogenic *Hras* in an in vitro model. We further investigated *miR-203* expression with in situ hybridization during skin tumorigenesis in vivo. In classic chemical carcinogenesis models initiated by DMBA/TPA treatment of mouse skin, *Hras* is preferentially mutated, which compromises the GTPase activity and results in constitutive Hras activation. Mutating *Hras* at Q61 leads to papilloma development and infrequent malignant transformation to SCCs (*Abel et al., 2009*). We first examined *miR-203* expression in benign papillomas. Consistent with our in vitro results, *miR-203* expression was absent from epithelial compartments adjacent to tumor stroma, the region where putative cancer stem cells reside. Although we also observed moderately expressed *miR-203* in tumor regions with evidence of cellular differentiation (keratin pearls and large differentiated morphology), the levels of *miR-203* was considerably lower in these regions compared to the suprabasal cells of adjacent normal skin, where *miR-203* is normally expressed (*Figure 2E*). Overall, *miR-203* expression levels were strongly reduced in tumor tissues compared to adjacent epidermal regions. In an independent mouse model of SCC, we also found that *miR-203* expression was gradually lost in a $Kras^{G12D}/Smad4^{cKO}$ model where skin tumors progressed to invasive SCC through serial passages of tumors (*White et al., 2013*) (*Figure 2F*). Taken together, these results demonstrate that *miR-203* is significantly downregulated at both early and late stages of papilloma and SCC formation in mouse models.

To evaluate the relevance of the loss of *miR-203* in human skin cancer, we examined 9 tumor samples obtained from patients with the early, middle, and late stages of skin cancer. In these human skin SCC samples, *miR-203* was already downregulated at the onset of tumorigenesis and

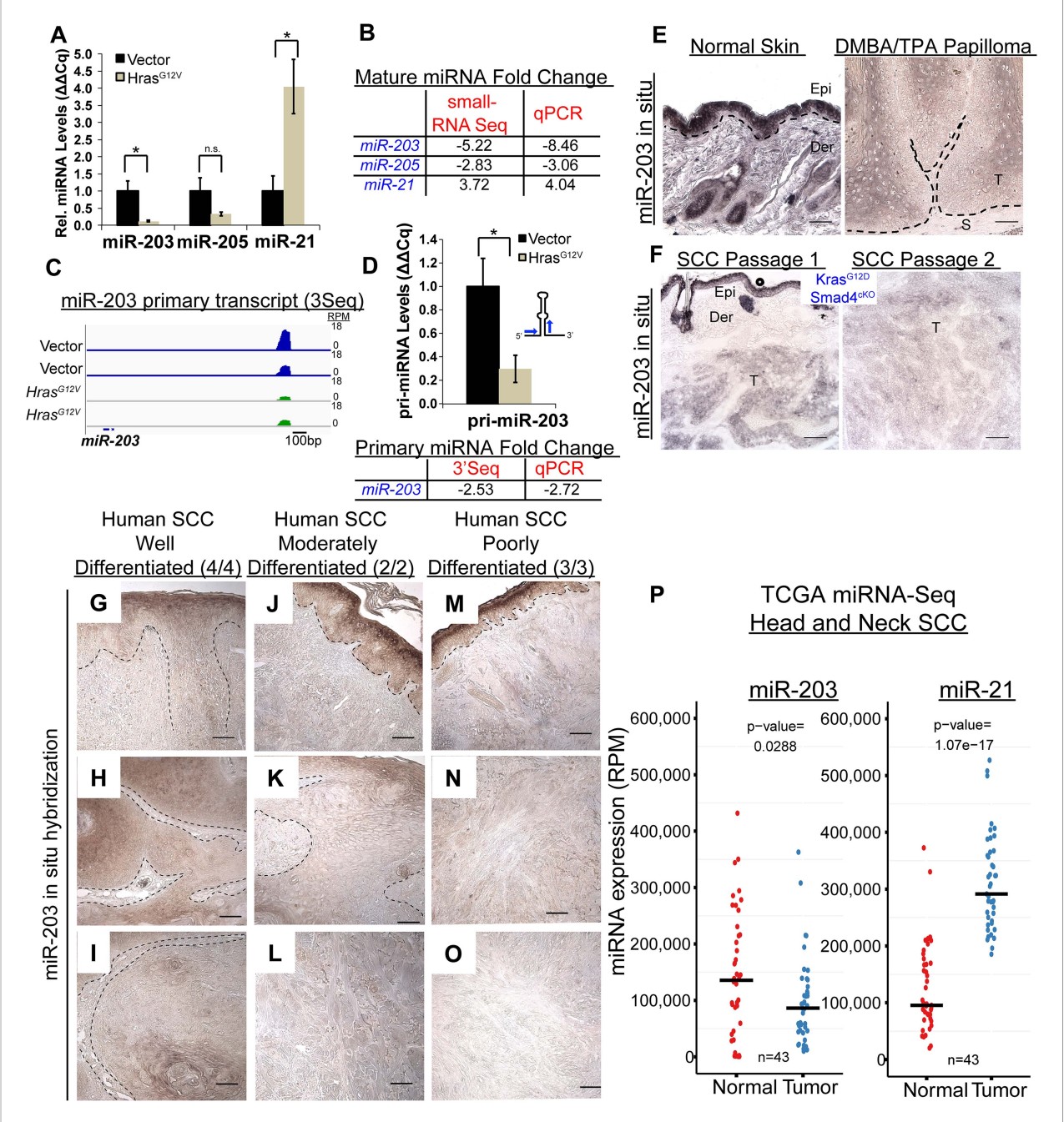

**Figure 2**. *miR-203* is strongly suppressed in mouse and human SCCs. (**A**, **B**) qPCR and small-RNA-Seq independently validate downregulation of miR-203 and upregulation of miR-21 driven by oncogenic *Hras^G12V* (n = 3 biological rep. qPCR, n = 2 small-RNA-Seq, mean ± SEM displayed, *p < 0.05, Student's *t*-test two-sided). (**C**) Gene track and quantification the 3'end of the *miR-203* primary transcript based on 3Seq. (**D**) *miR-203* primary transcript detection by qPCR (n = 3 biological replicates, Mean ± SEM displayed, *p < 0.05, ns = non-significant, Student's *t*-test two-sided). (**E**) *miR-203* is downregulated in DMBA/TPA produced papillomas compared to normal adjacent tissue. Epi = epidermis, Der = dermis, T = tumor, and S = stroma. The black lines denote the epidermal/dermal and tumor/stroma boundary (**F**) *miR-203* is downregulated in malignant SCCs derived from *Kras^G12D/Smad4^cKO* and passaged in immunocompromised mice. (**G**–**O**) Reduced *miR-203* expression is correlated with increasing malignancy in human skin SCC cancers. Panels **G**, **J**, **M** were taken from regions with more histologically normal regions to demonstrate successful *miR-203* hybridization. (**P**) miRNA-Seq quantification from patient matched normal and tumor tissue obtained from the TCGA consortium data (bar indicates mean value, Student's *t*-test two-sided). Scale bar = 50 µm.

progressively lost during the course of tumor progression, similar to the pattern observed in the mouse models. In the poorly differentiated SCC samples, the *miR-203* signal was completely absent, yet readily detectable in surrounding hyperplastic or normal epidermal regions (*Figure 2G–O*). In addition, we mined publically available data sets from The Cancer Genome Atlas (TCGA) and detected a significant reduction in *miR-203* and an elevation of *miR-21* in head and neck SCC samples compared to patient-matched normal tissues (*Figure 2P*). Altogether, these results corroborate our observations in mouse models and validate the strong correlation between the loss of *miR-203* and the development of skin cancer at multiple stages of tumorigenesis. These expression analyses suggest that the loss of *miR-203* coincides with the tumor-initiating events and *miR-203* might function as a tumor-suppressing mechanism in skin cancer.

## Genetic deletion of *miR-203* impacts early epidermal development in mouse

We then generated a conditional KO mouse model to assess the function of *miR-203* in murine skin development and carcinogenesis in vivo. We have characterized the genomic locus of *miR-203* and determined that *miR-203* is located in an intergenic region, 3.3 kbp downstream from the *Asp* gene and 15.3 kbp upstream of the *Kif26a* gene (*Figure 3—figure supplement 1*). Two loxP sites were inserted to flank the *miR-203* hairpin (*Figure 3A*). *miR-203* was deleted by first mating *miR-203^fl/fl* mice with *Actb-Flpe* mice to remove the *Neo* cassette, followed by breeding with *EIIa-Cre* or *Krt14-Cre* mice, resulting in complete *miR-203* loss from all tissues or only from skin tissues, respectively. In both cases, ablation of *miR-203* was confirmed by qPCR on isolated epidermis and in situ hybridization (*Figure 3B,C*). We previously demonstrated that expression of *miR-203* was largely restricted to stratified epithelial tissues. Within the skin, the differentiated skin lineages express *miR-203* ~10-fold higher than the stem/progenitor lineages. Consistent with this observation, there are no discernible differences between constitutive *miR-203* null (*miR-203^−/−*) mice and skin-specific cKO (*Krt14-Cre/miR-203^fl/fl*) mice. Both strains were born at the expected Mendelian ratio. They also showed no signs of gross developmental defects as adults (*Figure 3D*). For all subsequent studies, we used *miR-203^−/−*, which was maintained in the C57BL/6 background.

*miR-203* is highly expressed primarily in stratified epithelial tissues, such as the epidermis, tongue, esophagus, and cervix, yet poorly expressed or absent in other tissues such as the small intestine, bladder, lung, kidney, liver, or brain (*Figure 3—figure supplement 2*). Because *miR-203* begins to be expressed by E13 when the epidermis begins to stratify, we examined the proliferation rate and thickness of embryonic skin from E16 to P4. At E16, we observed a mild increase in cell proliferation in the KO, as measured by BrdU incorporation (*Figure 3E,F*). Although the difference in BrdU incorporation did not achieve statistical significance (p = 0.07), the thickness of the KO epidermis was significantly increased, compared to WT littermates (p = 0.03). Interestingly, the increase in epidermal thickness was most prominent at early stages (E16 and E17) and waned as skin development progressed (P4) (*Figure 3G*). At P4, *miR-203^−/−* mice displayed normal histological development of the epidermis and hair follicles, and the difference in proliferation and epidermal thickness between KO and WT became indistinguishable (*Figure 3G*). In addition, we found no evidence of perturbed differentiation in *miR-203^−/−* mice based upon analysis of early and late epidermal differentiation markers, *Keratin 1* and *Loricrin*, for the spinous and granular layers, respectively (*Figure 3H,I*). Together, these results provide evidence that *miR-203* limits cell division when the proliferation rate is high during early embryonic skin development but not at later stages when the proliferation rate wanes.

We noted that the impact of *miR-203* loss was correlated with the rate of cell proliferation. To further test whether *miR-203* functions to restrict the expansion of highly proliferative cells, we investigated the roles for *miR-203* in regulation of primary and established keratinocyte cell lines, respectively. We observed ~two-fold higher colony-forming capacity of *miR-203^−/−* keratinocytes, compared to the WT controls (*Figure 3J*). To further confirm the ability of *miR-203* to suppress cell proliferation cell-autonomously, we generated a *miR-203^fl/fl* keratinocyte cell line. By treating these cells with an adenoviral vector to express *Cre* (*Ad-Cre*), we determined that within 48 hr of *Ad-Cre* exposure *miR-203* was completely depleted (<1% remaining as measured by qPCR). We again observed a similar, ~two-fold higher colony-forming capacity by the *Ad-Cre*-treated *miR-203^fl/fl* cells compared to the *Ad-GFP*-treated control cells (*Figure 3K*). In both cases, although we detected some

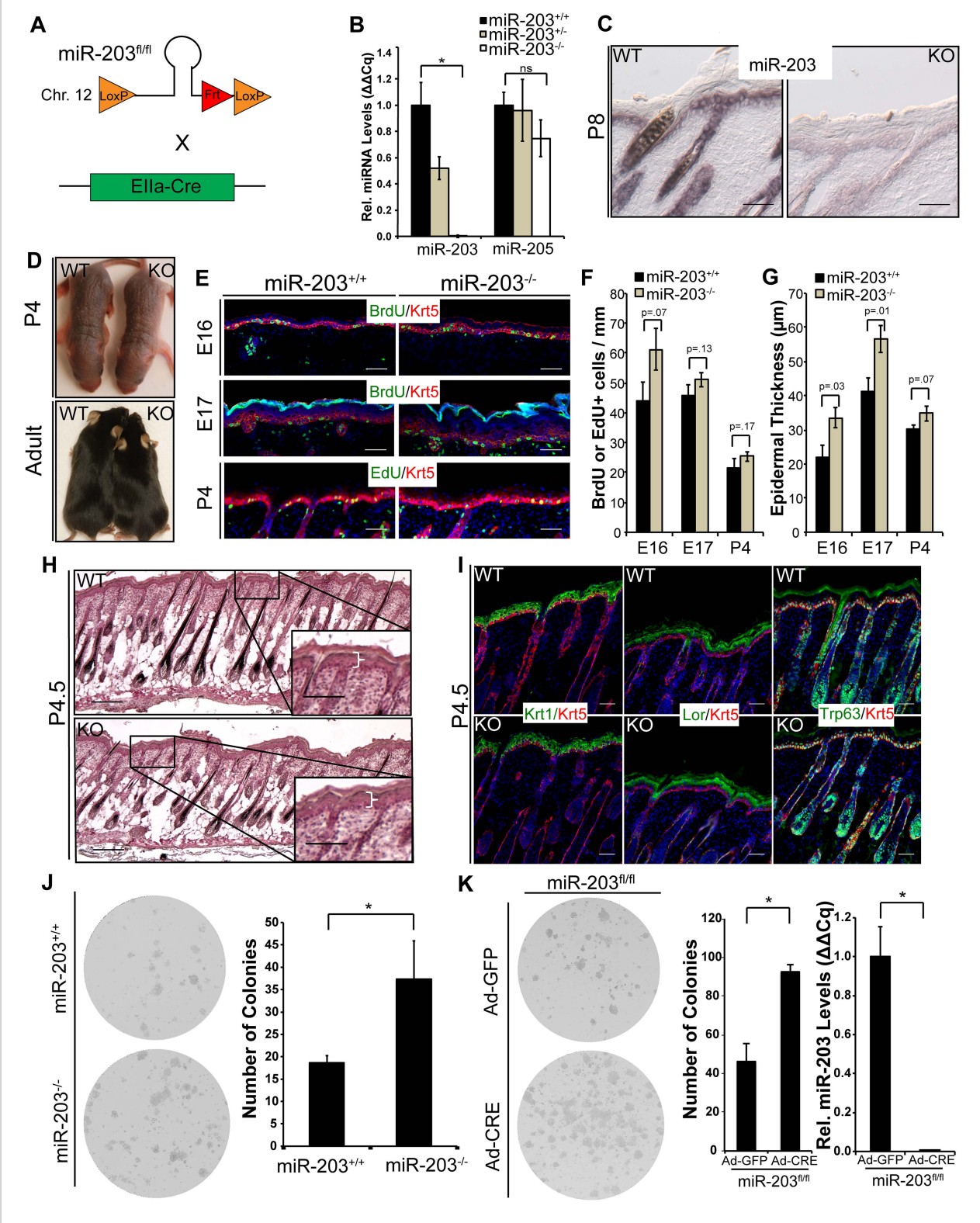

**Figure 3**. Loss of *miR-203* modestly impairs embryonic epidermal development. (**A**) Schematic of *miR-203* conditional allele generation and knockout strategy. (**B**) Validation of *miR-203* ablation by qPCR from isolated epidermal samples (n = 3 biological replicates, * p < 0.05, ns = non-significant, Student's *t*-test two-sided). (**C**) Validation of *miR-203* ablation within the epidermis by in situ hybridization (Scale bar = 50 μm). (**D**) *miR-203* knockout mice are visibly indistinguishable from wild-type counterparts. (**E–G**) *miR-203* ablation results in mild epidermal hyperplasia during embryonic development.
*Figure 3. continued on next page*

Figure 3. Continued

(n = 3 E16, n = 4 E17, and n = 3 p4 animals, p-value provided in figure, Student's *t*-test one-sided). (**H**) Representative hematotoxylin and eosin image from p4.5 animals, demonstrating restored normal skin morphology in neonatal animals. (Scale bars = 50 μm for inset and 100 μm for main images) (**I**) Epidermal differentiation is not compromised by loss of *miR-203*. (Scale bars = 50 μm) (**J**) *miR-203⁻/⁻* primary keratinocytes are more clonogenic than wild-type counterparts (representative results from 3 experiments, \*p < 0.05, Student's *t*-test two-sided). (**K**) Conditional ablation of *miR-203* from passaged *miR-203^{fl/fl}* keratinocytes results in higher clonogenicity (representative results from n = 3 independent experiments, mean ± standard deviation displayed, \*p < 0.05, Student's *t*-test, two-sided).

The following figure supplements are available for figure 3:

**Figure supplement 1**. Generation of a *miR-203* conditional knockout mouse.
**Figure supplement 2**. *miR-203* expression in diverse mouse tissues.

slightly larger clones with more cells formed by the *miR-203* null keratinocytes, the biggest differences were the significantly increased number of clones formed by the *miR-203* null cells. These results suggest that *miR-203* limits the clonogenicity of normal keratinocytes. When *miR-203* is deleted, more cells are likely to become clonogenic.

## Loss of *miR-203* enhances chemical carcinogenesis in the skin

We determined that the loss of *miR-203* is an early event in the initiation and development of both mouse and human skin SCCs. Furthermore, genetic deletion of *miR-203* confers ~two-fold higher colony-forming ability on primary keratinocytes. Because oncogenic *Hras* is a potent driver for tumorigenesis in the skin and our miR-seq data revealed a rapid and strong repression of *miR-203* by *Hras^{G12V}*, these observations prompted us to investigate whether the loss of *miR-203* plays a role in skin carcinogenesis. We subjected WT and *miR-203* null mice to two-stage chemical DMBA/TPA carcinogenesis (*Figure 4A*). The chemical carcinogenesis experiments were terminated at week 21 when tumor burden had reached a maximum. Because our mice were generated in the C57BL/6 background and these mice are known to be resistant to two-stage chemical carcinogenesis (*Abel et al., 2009*), all tumors generated in our mice were papillomas with no evidence for malignant conversion to squamous cell carcinomas during the time frame of our study. Nevertheless, we examined the temporal and numeric characteristics of tumor formation in our mice for the role of *miR-203* in tumor initiation. During the course of carcinogenesis, we observed a slightly earlier tumor formation pattern on the backskin of *miR-203* null animals (*Figure 4B*). Furthermore, *miR-203* null animals developed ~2.5-fold more tumors, when compared to WT control animals (*Figure 4B*). Measurement of tumor sizes at week 17 showed no statistically significant difference in tumor sizes between genotypes although *miR-203* null animals were more susceptible to tumor formation (*Figure 4C*). These results suggest that loss of *miR-203* increases the number of tumor-initiating cells but does not significantly alter the type of tumors for example, conferring tumors with more aggressive phenotypes.

Hematoxylin and Eosin staining revealed that the papillomas produced had similar morphology, displaying exophytic lesions with evidence for squamous differentiation (*Figure 4D,E*). Assessment of proliferation (*Ki67*), differentiation markers (*Krt1, Lor*) revealed similar dynamics between *miR-203^{+/+}* and *miR-203^{-/-}* tumors (*Figure 4F*). To further probe the mechanistic differences between *miR-203* WT and KO tumors, we assessed the mutation spectra of these tumors and found that they possess the canonical *Hras^{Q61L}* mutation at similar frequencies, 80% and 85.7% for WT and KO tumors, respectively (*Figure 4—figure supplement 1*). Taken together, these results provide direct evidence for *miR-203*'s tumor suppressing roles at the stage of tumor initiation in the classic DMBA/TPA tumorigenic model.

## *miR-203* represses clonal selection of oncogenic *Hras*-transformed cells in vitro

To further probe *miR-203*'s role in clonal selection, we infected *miR-203* WT and null primary keratinocytes with *pBabe-Hras^{G12V}*. At the passage 1, the loss of *miR-203* led to ~40% increase in

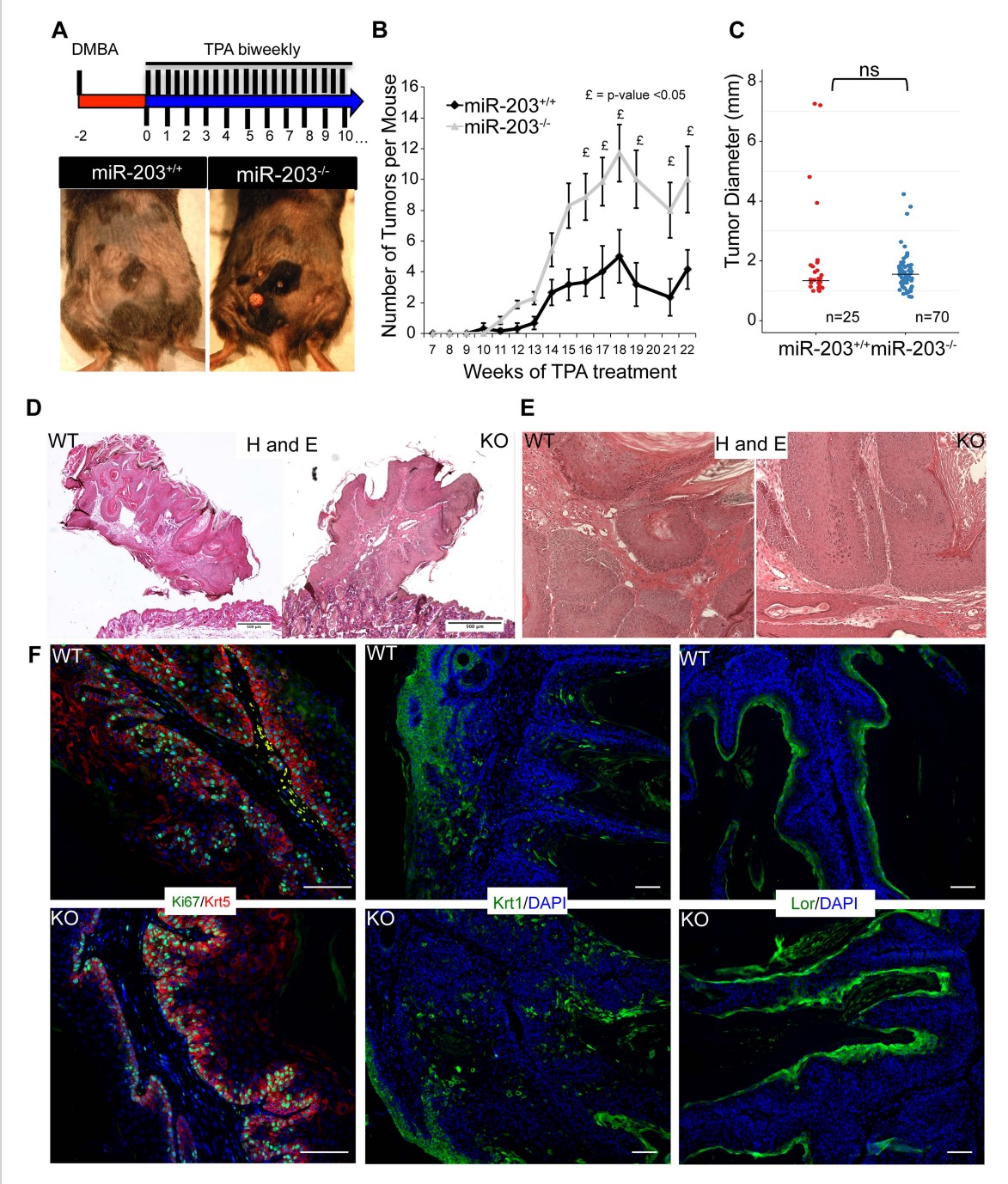

**Figure 4**. Loss of *miR-203* sensitizes mice to DMBA/TPA skin carcinogenesis. (**A**) Representative images of tumors that were formed in the skin of WT and *miR-203* null mice treated with DMBA/TPA. (**B**) *miR-203⁻/⁻* mice have a larger tumor burden than *miR-203⁺/⁺* counterparts (n = 6 and 7 *miR-203⁺/⁺* and *miR-203⁻/⁻* animals respectively, mean ± SEM displayed, £ = p < 0.05, Whitney–Mann U-test one-sided). (**C**) *miR-203⁻/⁻* tumor size distribution is similar to wild-type animals (ns = non-significant, Student's *t*-test two sided, median displayed as bar). (**D**, **E**) *miR-203⁺/⁺* and *miR-203⁻/⁻* papillomas display similar morphologies and histology. (**F**) Proliferation and differentiation dynamics are similar between *miR-203⁺/⁺* and *miR-203⁻/⁻* tumors. (Scale bars = 50 μm).

The following figure supplement is available for figure 4:

**Figure supplement 1**. The *Hras^Q61L* mutation is common in both *miR-203⁺/⁺* and *miR-203⁻/⁻* tumors.

colony-forming capacity immediately after the initial plating (*Figure 5A*). When we passaged the *Hras*[G12V] transduced cells, we began to observe reduced numbers of colonies that were formed by both the WT and null cells (*Figure 5A*). This was likely due to oncogenic stress caused by *Hras*[G12V] induction. Strikingly, in contrast to the WT cells that generally formed smaller colonies in subsequent passages, the *miR-203* null cells generated more and bigger clones compared to WT control cells at each passage (*Figure 5A*). Collectively, the serial passage assay supported the enhanced selection of tumor-initiating cells in the absence of *miR-203* upon oncogenic *Hras* induction.

To further characterize the ability of *miR-203* to suppress the growth of oncogenic *Hras*-transformed cells, we used our previously established *Krt14-rtTA/pTre2-miR-203*-inducible model (*Jackson et al., 2013*). After infecting the inducible keratinocytes with either the *pBabe* vector control or *pBabe-Hras*[G12V], we treated the cells with 5 µg/ml doxycycline to induce ~4- to 7-fold increase in *miR-203* expression, a physiologically relevant level of *miR-203* typically observed during epidermal differentiation (*Figure 5B*). Introduction of *miR-203* resulted in suppression of keratinocyte proliferation and colony formation ability (*Figure 5C,D*), as noted previously (*Yi et al., 2008*; *Jackson et al., 2013*; *Benaich et al., 2014*). Furthermore, whereas oncogenic *Hras*-enhanced S-phase entry, short-term (~24 hr) induction of *miR-203* completely abolished the gain of S-phase entry (*Figure 5D*). Over a longer term, continuous induction of *miR-203* severely compromised the colony-forming capacity of the transduced cells (*Figure 5C*). We did not detect any evidence for enhanced apoptosis caused by *miR-203*, as measured by the absence of sub-G1 keratinocytes. Taken together, our results suggest that the loss of *miR-203* is critical for the initial selection and expansion of primary keratinocytes harboring the oncogenic *Hras* mutation and the gain of *miR-203* can effectively suppress the growth of these cells.

## Comprehensive identification of *miR-203* targets reveals regulation of the Ras signaling pathway

Our data so far have suggested a role of *miR-203* in suppressing the expansion of tumor-initiating cells driven by oncogenic *Hras* mutations. To decode the underlying mechanism, we carried out comprehensive analyses to identify *miR-203* targets in the skin. Recent studies demonstrated that miRNAs' impact on gene expression could be largely captured by measuring the changes of mRNA levels upon manipulation of miRNA expression (*Guo et al., 2010*; *Eichhorn et al., 2014*). In parallel, the high-throughput sequencing of RNA isolated by crosslinking immunoprecipitation (HITS-CLIP) approach directly crosslinks miRISC to their targets and identifies miRNA targets through physical interaction (*Chi et al., 2009*). Therefore, we took a combinatorial approach integrating multiple data sets obtained from our KO and inducible mouse models with *Ago2* HITS-CLIP analysis.

We applied several different profiling techniques including ribosome profiling, RNA-Seq (3Seq) and microarray methods to determine upregulated genes in *miR-203* KO samples and downregulated genes in *miR-203*-induced samples. For ribosome profiling and 3Seq, we used primary keratinocytes, which are amenable to effective cycloheximide treatment and therefore also allowed us to investigate *miR-203*'s impact on translation efficiency (*Figure 6A*). For microarray analysis, we used freshly isolated epidermis obtained from *miR-203* WT and KO animals at P4.5. To interrogate the impact of *miR-203* gain-of-function, we used the *Krt14-rtTA/pTre2-miR-203*-inducible mouse model that allows us to control the duration and dosage of *miR-203* expression (*Jackson et al., 2013*). We applied microarray profiling to determine downregulated genes in FACS-purified neonatal epidermal and hair follicle progenitors with a short-term (24 hr) induction of *miR-203* (*Figure 6A*). Altogether, we accrued 20 genome-wide expression data sets from *miR-203* WT, KO and inducible samples. This comprehensive collection of functional genomic data allowed us to characterize the action of *miR-203* on the transcriptome.

We first analyzed the *miR-203* overexpression samples. GO-analysis demonstrated that genes involved in regulation of mitotic cell cycle, DNA synthesis, and metabolism were prominently downregulated upon *miR-203* induction (*Figure 6—source data 1*). *miR-203* upregulation in both epidermal and hair follicle progenitors resulted in strong suppression of transcripts harboring perfect seed matches to *miR-203* in their 3′UTRs (*Figure 6—figure supplement 1*). Among them, transcripts containing the 8-mer matches were strongly suppressed, compared to transcripts without a *miR-203* seed match (p = $1.5 \times 10^{-23}$ for epidermis and $2.6 \times 10^{-25}$ for hair follicle, respectively) (*Figure 6—figure supplement 1A*). In contrast to *miR-203* overexpression, deletion of *miR-203* did

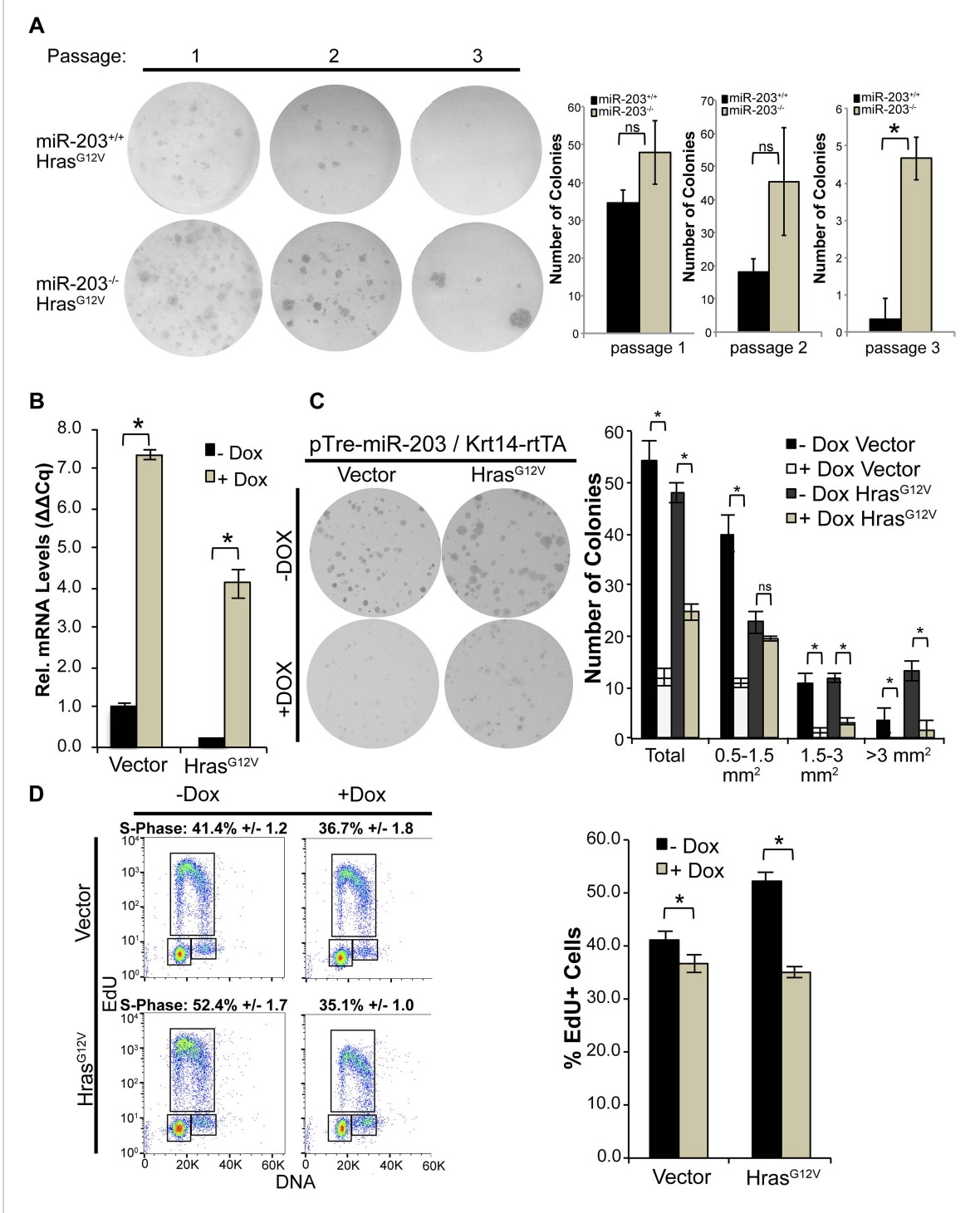

**Figure 5**. *miR-203* antagonizes *Hras^G12V*-driven keratinocyte proliferation. (**A**) *Hras^G12V* transduced *miR-203^−/−* primary cultures are more colonogenic upon serial passage than wild-type controls (representative of n = 2 independent experiments, mean ± standard deviation displayed. *p < 0.05, ns = non-significant, Student's *t*-test two-sided). (**B**) qPCR of *miR-203* induction upon addition of doxycycline in vector and *Hras^G12V* transduced cells (mean ± SEM displayed, n = 3 biological replicates). (**C**) Restoration of *miR-203* using a doxycycline-inducible transgene results in suppression of colony formation ability in *Hras^G12V* transduced and control keratinocytes. *miR-203* was induced with doxycycline (5 µg/ml) 24 hr after plating (representative of n = 3 independent experiments, mean ± standard deviation displayed, *p ≤ 0.05, , ns = non-significant). (**D**) *miR-203* restoration suppresses *Hras^G12V*-driven S-Phase entry. *miR-203* was induced for 24 hr prior to harvesting for flow cytometry. (n = 3, mean ± standard deviation displayed, *p ≤ 0.05).

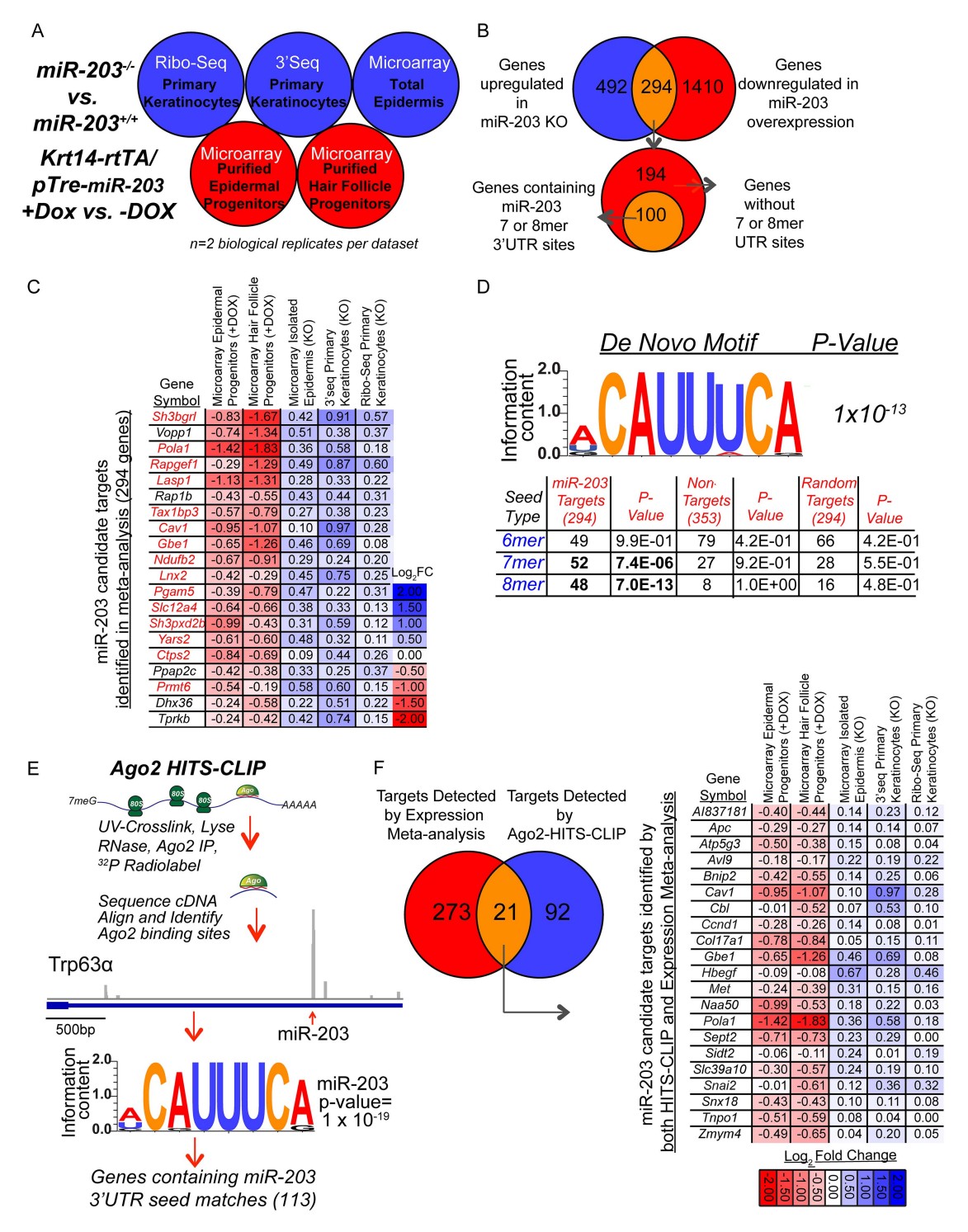

**Figure 6**. Comprehensive identification of *miR-203* targets using genome-wide expression analyses and Ago2 HITS-CLIP. (**A**) Schematic of genome-wide expression profiling data sets used in meta-analysis to identify *bono-fide miR-203* targets. (**B**) Genes upregulated in all three *miR-203* loss-of-function data sets (786 genes, no fold-change or p-value cut-off) were compared to genes downregulated in both *miR-203* gain-of-function data sets (1704 genes, no fold-change or p-value cut-off) to identify a subset of genes with a strong inverse correlation to *miR-203* expression (294 genes) of which 100 genes contained *miR-203* 7mer or 8mer seed sequence matches in their 3'UTRs. (**C**) Table demonstrating top 20 genes identified in meta-analysis ranked by negative-correlation to *miR-203* expression. Genes colored in red contain 3' UTR *miR-203* 7 or 8mer seed matches. (**D**) De novo motif searching identified an 8mer *miR-203* seed motif, complementary to the *miR-203* seed sequence enriched in the 3'UTR of candidate *miR-203* target genes identified in the

*Figure 6. continued on next page*

*Figure 6. Continued*

meta-analysis (294 genes). Table demonstrating enrichment for 7 or 8mer seed matches in the 3′UTR of candidate *miR-203* target genes (294) over the background seed distribution in primary keratinocytes, which is not seen for randomly selected 294 genes expressed in primary keratinocytes or a negative control gene set of genes upregulated in *miR-203* gain-of-function and downregulated *miR-203* loss-of-function (353 genes). (**E**) Schematic of *Ago2* HITS-CLIP and the identified *miR-203* seed motif. (**F**) Diagram of genes detected by expression meta-analysis and *Ago2*-HITS-CLIP. Table of 21 high confidence *miR-203* targets identified through expression meta-analysis and that have *Ago2*-HITS-CLIP 3′UTR peaks with miR-203 seed matches.

The following source data and figure supplements are available for figure 6:

**Source data 1**. GO-analysis of selected *miR-203* data sets.

**Source data 2**. Putative *miR-203* targets detected in the expression meta-analysis.

**Figure supplement 1**. Transcripts containing 3′UTR *miR-203* seed matches are regulated by *miR-203*.

**Figure supplement 2**. *Ago2*-HITS-CLIP in primary keratinocytes.

**Figure supplement 3**. *Ago2* HITS-CLIP 3′UTR peaks are enriched in keratinocyte miRNA seed matches, including *miR-203*.

**Figure supplement 4**. Predicted *miR-203* targets based on HITS-CLIP are regulated by *miR-203*.

**Figure supplement 5**. *miR-203* targets do not display translation efficiency changes upon *miR-203* ablation.

not perturb transcript levels genome-wide as strongly as observed under the induced conditions and offered little insight into miR-203's function based upon GO analysis (*Figure 6—source data 1*). However, despite the relatively mild changes in the transcript levels in the KO skin and primary keratinocytes, transcripts that contain 8-mer 3′UTR matches were still more significantly upregulated than transcripts without any *miR-203* seed matches ($p < 0.05$) (*Figure 6—figure supplement 1B*). Lastly, a ranked sum analysis that ranked genes based on expression changes correlated with the levels of *miR-203* showed that genes with the 8-mer or 7-mer seed ranked significantly higher than those transcripts without seed matches (*Figure 6—figure supplement 1C*).

By demanding all potential targets be upregulated in all the *miR-203* KO samples and downregulated in all the *miR-203*-induced samples, we identified 294 transcripts as our candidates for *miR-203* targets (*Figure 6B*). Among the top 20 most differentially expressed transcripts when *miR-203* was deleted or induced, 15 of them contain at least one 7-mer or 8-mer match in their 3′UTRs (*Figure 6C* and *Figure 6—source data 2*). In support of this approach, an unbiased de novo motif search using the 3′UTRs of these 294 transcripts revealed that the most enriched motif (ACAUUUCA, $p = 1 \times 10^{-13}$) perfectly matched to nucleotide position 2–9 of *miR-203*, the 8-mer seed sequences of *miR-203* (*Figure 6D*). Additional statistical analysis of the number of genes containing perfect 3′UTR matches to the 7-mer or 8-mer seed among these candidates revealed significant enrichment over the background distribution among mRNAs expressed in the skin (*Figure 6D*). Therefore, we selected genes containing 7-mer or 8-mer seed matched 3′UTR sites as *miR-203* target candidates and obtained a collection of 100 potential *miR-203* targets (*Figure 6B*).

We then applied HITS-CLIP to interrogate the direct interaction between *miR-203* and its mRNA targets. We generated four *Ago2* HITS-CLIP libraries from primary WT keratinocytes, which abundantly express *miR-203* (*Figure 1G*). *Ago2*-RNA complexes were isolated from a region extending from approximately 110 kd–130 kd as expected (*Figure 6—figure supplement 2A*). Sequencing libraries from *Ago2* HITS-CLIP samples were generated and analyzed using previously described methods (*Chi et al., 2009*; *Moore et al., 2014*). The HITS-CLIP faithfully captured miRNA species expressed in keratinocytes, including *miR-203* (*Figure 6—figure supplement 2B,E*). Overall our HITS-CLIP reads and clusters alignment were similar to previous published results with a significant portion aligned to 3′UTRs (*Figure 6—figure supplement 2D*). To further validate our approach, we analyzed the positional enrichment of miRNA seed sequences within the 3′UTR HITS-CLIP clusters and detected strong enrichment over dinucleotide shuffled cluster sequences or randomly distributed 3′UTR regions (*Figure 6—figure supplement 3A*). Additionally, de novo motif searching for 8mer

motifs identified the seed sequences of miRNAs that are highly expressed in keratinocytes (*Figure 6—figure supplement 3B*). The motif corresponding to *miR-203* detected by the HITS-CLIP was also ACAUUUCA (p = 1 × 10⁻¹⁹), identical to the motif detected by our transcriptome analysis (*Figure 6E* and *Figure 6—figure supplement 3B*). A total of 113 mRNAs were detected to have *miR-203* seed containing *Ago2* binding sites. We next examined the ranked sum expression of these targets and determined that transcripts with *Ago2* HITS-CLIP *miR-203* binding sites were ranked significantly higher compared to non-targeted mRNAs, indicating that many of these targets are functional (*Figure 6—figure supplement 4*). Finally, we did not detect any evidence for global regulation at the translation level for *miR-203* targets based on our analysis of translational efficiency changes upon *miR-203* deletion (*Figure 6—figure supplement 5*). Together, these HITS-CLIP data independently validated that *miR-203* uses its seed sequences for mRNA targeting and predicted 113 *miR-203* targets based on *Ago2* binding.

By combining the targets detected by our differential expression data sets and by the HITS-CLIP, we identified a list of high-confidence targets for *miR-203* (*Figure 6F*). Importantly, we found a number of key regulators in the Ras signaling pathway and important genes involved in regulation of cell division including *Hbegf*, *Ccnd1*, *Snai2*, *Met,* and *Pola1* in this list (*Figure 6F*). This collection of *miR-203* targets suggests that *miR-203* targets multiple pathways, including several components of the *Ras* signaling pathway, to suppress cell proliferation.

## *miR-203* directly targets *Hbegf* and *Pola1*

Our genome-wide analyses identified a number of novel targets of *miR-203*. Given our findings that the loss of *miR-203* promotes the selection and expansion of oncogenic *Hras*-transformed cells both in vivo and in vitro, we were interested in understanding the underlying mechanism. Overall, *miR-203* targets identified in the meta-analysis were enriched in the upregulated transcripts in *Hras*^G12V^-transformed keratinocytes, compared to non-targeted transcripts, consistent with the down-regulation of *miR-203* in these cells (*Figure 7—figure supplement 1A*). Among these targets, we were intrigued by the observation that multiple regulators of the *Ras* signaling pathway and critical factors for DNA replication and cell cycle progression were among our high confidence targets and additionally were among the most upregulated *miR-203* targets in *Hras*^G12V^-transformed keratinocytes (*Figure 7—figure supplement 1B*). To begin to validate the high-confidence set of *miR-203* targets, we selected *Pola1* and *Hbegf* for further study. *Pola1* is the catalytic subunit of the DNA-POL-alpha holoenzyme, which is required in initiation of DNA replication during S-phase (*Lehman and Kaguni, 1989*). *Hbegf* is an Egf-like ligand that activates MAPK signaling through activation of EGF-receptors, *Erbb1* and *Erbb4*. In keratinocytes, *Hbegf* is mitogenic and promotes keratinocyte migration (*Stoll et al., 2012*). In an epithelial cancer cell line, *Hbegf* acts as an oncogene promoting cell proliferation (*Miyamoto et al., 2004*). *Pola1* and *Hbegf* contain 3′ UTR *miR-203* target sites (9-mer and 8-mer respectively) that are targeted by *miR-203*, validated by luciferase assay (*Figure 7A*). Furthermore, in *miR-203* null epidermis, both *Pola1* and *Hbegf* mRNAs were elevated (*Figure 7B*). In addition to mRNA levels, *Pola1* protein levels were also elevated in the absence of *miR-203*. It was further elevated in the presence of *Hras*^G12V^ and repressed by *miR-203* induction (*Figure 7C,D*). We were unable to measure the protein level of *Hbegf* due to poor antibody performance. Additionally, we observed that the expression of *Ccnd1*, an essential cell cycle regulator that is often induced or amplified by oncogenic *Ras* (*Downward, 2003*; *Stransky et al., 2011*), showed a strong negative correlation to *miR-203* (*Figure 7C,D*). This suggested that loss of *miR-203* increases the levels of *Ccnd1* and contributes to the observed upregulation of this critical gene.

To assess the functional consequences of *Hbegf* and *Pola1* suppression on keratinocyte proliferation, we knocked down these targets using three independent shRNAs. Suppression of *Hbegf* and *Pola1* strongly suppressed the growth potential of keratinocytes (*Figure 7E*). Additionally, we knocked down *Hbegf* and *Pola1* in established *miR-203*^+/+^ and *miR-203*^−/−^ keratinocytes transduced with *Hras*^G12V^ and similarly observed potent suppression of keratinocyte growth potential, demonstrating that these targets are also required in *Hras*^G12V^-transformed keratinocytes (*Figure 7—figure supplement 2*). Taken together, these results validate *Hbegf* and *Pola1* as direct targets of *miR-203*. We hereby propose a model in which *miR-203* restricts selection and expansion of *Hras*-mutated cells by repressing multiple targets, a subset of which are involved in the *Ras* signaling pathway (*Figure 7F*).

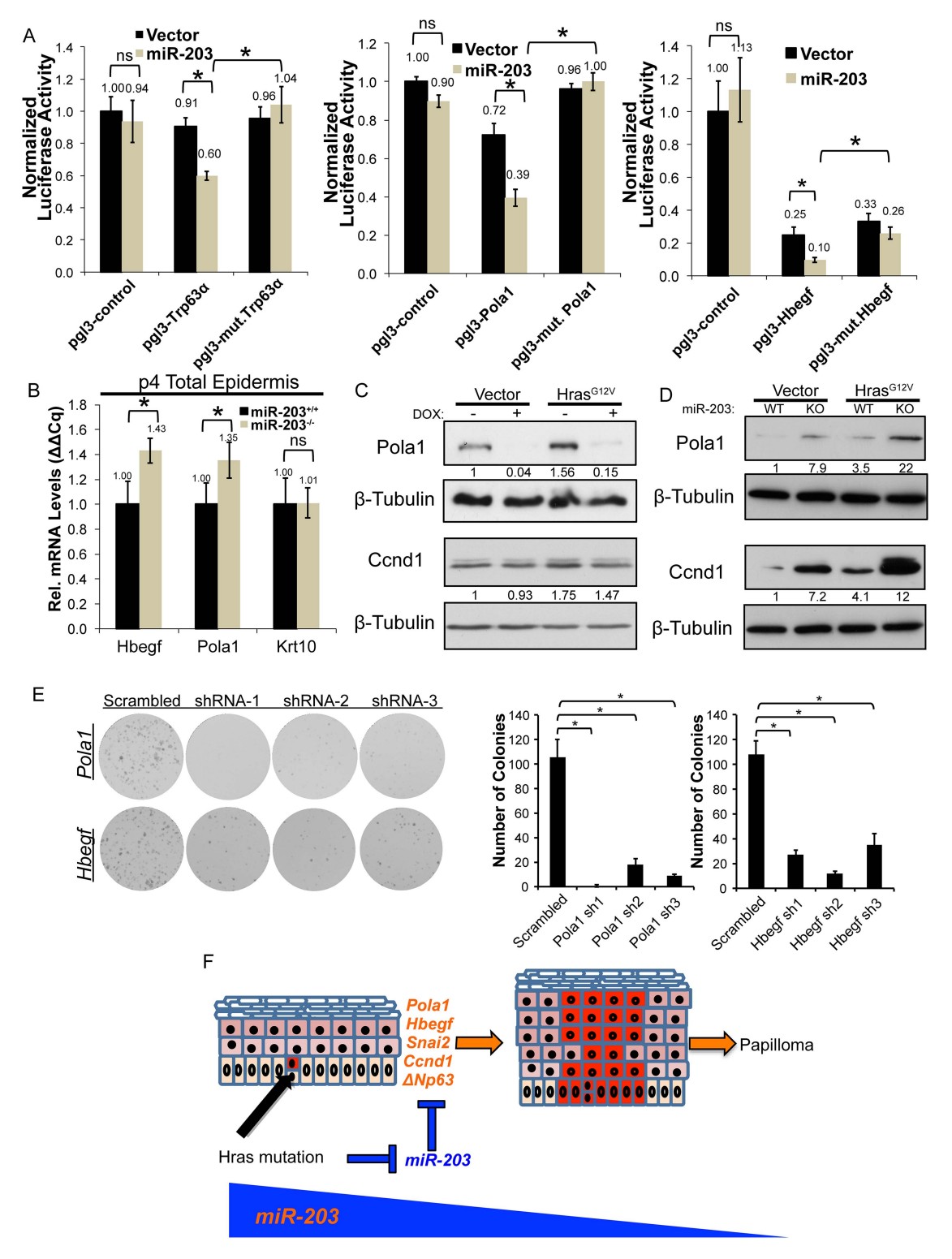

**Figure 7**. *Hbegf* and *Pola1* are direct *miR-203* target genes critical for keratinocyte proliferation. (**A**) 3'UTR luciferase reporter assays demonstrate that *miR-203* directly targets *Trp63* (positive control), *Pola1*, and *Hbegf* in keratinocytes (representative of n = 3 independent experiments, mean ± propagated standard deviation displayed, *p < 0.05, ns = non-significant, Student's *t*-test two-sided) (**B**) *Hbegf* and *Pola1* are upregulated in *miR-203^{−/−}* isolated epidermis (p4) (n = 8 and n = 10, *miR-203^{+/+}* and *miR-203^{−/−}* animals respectively, mean ± SEM displayed, *p < 0.05). (**C, D**) Western blots from lysates with *miR-203* overexpression (48 hr) or *miR-203* ablation. (**E**) shRNA knockdown of *Hbegf* or *Pola1* impairs keratinocyte colony formation ability (representative

*Figure 7. Continued*

of n = 3 independent experiments, *p <0.05, mean ± standard deviation displayed). (**F**) Model for the mechanism of *miR-203* in restricting *Hras*-initiated tumorigenesis.

The following figure supplements are available for figure 7:

**Figure supplement 1**. A subset of *miR-203* targets are upregulated by *Hras$^{G12V}$*.

**Figure supplement 2**. *Hbegf* and *Pola1* are required for keratinocyte growth potential in *Hras$^{G12V}$*-transformed *miR-203$^{+/+}$* and *miR-203$^{-/-}$* cultures.

## Discussion

Identification and experimental validation of driver gene mutations have been instrumental for our understanding of cancer biology (**Vogelstein et al., 2013**). Although it is clear that not all cells harboring driver gene mutations develop into tumors, the mechanism of clonal selection within cells that have acquired driver gene mutations remains poorly understood. In this study, we used an oncogenic *Hras*-induced tumor model to determine the molecular consequences of this oncogenic driver mutation and to study the selection process. We quantitatively measured the impact of oncogenic *Hras$^{G12V}$* on the landscape of mRNA and miRNA expression. Whereas our mRNA-Seq data confirmed the profound ability of *Hras$^{G12V}$* in promoting cellular transformation, our miR-Seq data revealed new insights into the dynamic changes in highly expressed miRNAs caused by *Hras$^{G12V}$* (**Figure 1**). In particular, the downregulation of *miR-203*, one of the most highly expressed miRNAs in the skin, likely occurs at the transcriptional level (**Figure 2C,D**). By examining tumor samples obtained from both mouse and human at multiple stages, we showed that the downregulation of *miR-203* is associated with tumor initiation and progression (**Figure 2E–P**). To determine the role of *miR-203* in this process, we generated a KO mouse model. Initial insights into the function of *miR-203* came from the analysis of *miR-203* null epidermis during skin development. Whereas we and others have shown the potent inhibition of epidermal proliferation by *miR-203* with gain-of-function approaches (**Lena et al., 2008**; **Yi et al., 2008**; **Jackson et al., 2013**; **Benaich et al., 2014**), it is unknown how the loss of *miR-203* affects the skin and mouse development in general. By analyzing the KO mouse, we revealed an early, albeit mild, hyperproliferative phenotype in the embryonic but not postnatal skin (**Figure 3E–I**). However, when we subjected *miR-203* WT and null cells obtained from postnatal skin for their colony-forming ability, we observed an increased ability of the *miR-203* null cells to give rise to colony-forming cells (**Figure 3J,K**). This observation was indicative of a role for *miR-203* in restricting the expansion of highly proliferative cells. Of note, we did not observe any defects in the formation of the differentiated layers of the skin in the *miR-203* KO mice. This suggests that the primary function of *miR-203* is to restrict cell proliferation but not to promote epidermal differentiation per se. When we examined the role of *miR-203* during DMBA/TPA-mediated chemical carcinogenesis, we observed strong increase in the number of papillomas formed in the KO skin (**Figure 4A–C**). In vitro colony formation assays with serial passages further illustrated the enhanced ability for selecting tumor-initiating cells with *miR-203* KO cells when subjected to *Hras$^{G12V}$* transformation (**Figure 5A**). However, due to tumor resistance of our C57BL/6 strain, we were unable to determine if the loss of *miR-203* also promotes malignant transition. Of note, a recent study showed interesting results that restoration of *miR-203* suppresses human SCC metastasis (**Benaich et al., 2014**). Collectively, our results provide experimental evidence for an important role of *miR-203* in suppressing clonal selection and expansion of *Hras*-mutated nascent cancer cells.

A major challenge in understanding miRNA functions is to identify their high-confidence targets globally. To this end, we employed two independent approaches: transcriptome analysis with our KO and inducible models and direct miRNA target capture by *Ago2* HITS-CLIP (**Figure 6**). Importantly, with our expansive data set, we confirmed that *miR-203* utilizes the 5′ seed sequences to target perfectly matched sequences located on the 3′UTRs of its target genes. Consistent with recent genome-wide studies for the impact of miRNA on mRNA levels and translation efficiency in mammals (**Guo et al., 2010**; **Eichhorn et al., 2014**), we found no evidence for global changes of translational efficiency in the absence of *miR-203*, one of the most highly expressed miRNAs in the skin. However, we also noted that *Trp63*, a well-established *miR-203* target, showed little change at the mRNA levels

under all conditions (not shown). Our analysis with differential gene expression did not identify *Trp63* as a *miR-203* target. In contrast, the recognition of the 3′UTR site of *Trp63* by *miR-203* was robustly detected by the HITS-CLIP (*Figure 6E*). Thus, it is likely that our identification of *miR-203* targets based on the transcriptome analysis alone was conservative. More studies will be required to further integrate direct miRNA target capture with transcriptome analysis for miRNA target identification. Nevertheless, we still observed strong enrichment of *miR-203* targets in genes required for cell proliferation. These high-confidence targets illustrate the broad targeting of cell proliferation by *miR-203* and suggest mechanisms by which loss of *miR-203* can facilitate the expansion of oncogenic *Hras*-transformed cells. These new insights point to the potential of utilizing *miR-203* to simultaneously target multiple effectors required for cell division in human cancers. We also note that *miR-203* is known to antagonize human papillomaviruses (HPV) (*Melar-New and Laimins, 2010*). Our identification of *Pola1*, a critical cellular gene required for HPV genome replication, as a target of *miR-203* may provide a potential mechanism for miR-203-mediated antagonism against HPV infection. Finally, the relatively mild requirement for *miR-203* in slowly proliferating cells suggests that a reduction or increase in the level of *miR-203* may be well tolerated by most normal cells. This unexpected discovery could be further explored to determine *miR-203*'s therapeutic potential in treating certain types of epithelial cancers.

## Materials and methods

### 3seq library construction

3′seq libraries were constructed using methods described previously (*Wang et al., 2013*). Briefly, 500 ng of total RNA isolated from primary keratinocytes was poly-A purified (Dynabeads, Thermo Fisher Scientific, Waltham, MA) and chemically fragmented by treatment 95°C for 8 min in Fragmentation Buffer (Thermo Fisher Scientific). Fragmented RNAs were then oligoDT primed with a P7T20V oligo and reverse transcribed using SuperScript III (Thermo Fisher Scientific) (see *Supplementary file 2* for Oligo Sequences). Following ethanol precipitation, ligation competent cDNAs were generated via second-strand synthesis using RNase H and DNA Pol I enzymes, end-repaired using a mix of T4 DNA polymerase, Klenow DNA polymerase and T4 Polynucleotide Kinase in the presence of dNTPs (New England Biolabs, Ipswich, MA), and A-Tailed using Klenow 3′ to 5′ exo- in the presence of dATP (New England Biolabs). Ligation was then performed using the P5 Adaptor with T4 DNA Ligase for 1 hr (New England Biolabs). cDNA inserts 80–100 nt in length were isolated from 8% PAGE gels and subject to PCR amplification using RP1 and RT-Primer oligos for 12–16 cycles (Phusion, New England Biolabs). PCR products were isolated from PAGE gels and subject to 6 cycles of secondary PCR to introduce unique indices for multiplexed DNA sequencing on a HiSeq 2000 (Illumina, San Diego, CA).

Bioinformatics processing of 3seq data was performed as described previously, including adaptor trimming, read alignments, peak calling, peak filtering to remove internal polyA priming events, and transcript quantification with minor modifications (*Wang et al., 2013*). Following read alignment to the mm10 genome, alignments from every library were grouped together for peak calling and 3′end filtering to create a master set of high confidence 3′ end peaks. From this 3′end data set, reads counts for each library were obtained (see *Supplementary file 1* for mapping statistics). To obtain transcript read counts, reads from all peaks that passed our internal priming filter and were 3′UTR localized were summed to obtain an overall transcript count for each transcript. Normalized transcript counts were calculated as reads per million reads mapped (RPM). To determine differential transcript expression, low-abundance transcripts with less than 10 reads in two libraries were first excluded then the remaining transcripts were analyzed with EdgeR with classical analysis parameters.

### Small RNA-Seq

Small RNA-Sequencing libraries were prepared using minor modifications of a previously published protocol that reduces RNA-ligation biases and enables accurate miRNA quantification (*Zhang et al., 2013*). Briefly, two micrograms of total RNA was first ligated to a pre-adenylated 3′ linker (10 μM) using truncated T4 RNA Ligase 2 (20 Units New England Biolabs) in the presence of PEG-8000 (10% wt/vol) RNaseOUT (20 Units Thermo Fisher Scientific) for 4 hr at room temperature. Ligation reactions were then resolved on 7 M Urea-15% PAGE gel to isolate the miRNA-3′ adaptor hybrids. Following overnight elution from the acrylamide gel in HSCB buffer (400 mM NaCl, 25 mM Tris–HCl pH 7.5, 0.1% SDS), ligation products were ethanol precipitated, resuspended in 5′ ligation reaction mixture containing a 5′ linker (10 μM), 1 mM ATP, Peg-8000 (20% wt/vol), 1 × T4 RNA Ligase buffer (New

England Biolabs), denatured for 5 min at 70°C, after which RNaseOUT (20 units) and T4 RNA Ligase 1 (10 units) were added. Following ligation for 37°C for 2 hr, cDNA was generated via reverse transcription using Superscript III (Thermo Fisher Scientific) in the presence of a 3′adaptor specific primer. cDNA products were then subjected to 10–14 PCR cycles and resolved on 8% native PAGE gels. Libraries were then eluted as described above, precipitated, and submitted for sequencing on a HiSeq 2000 (Illumina).

Small-RNA sequencing reads were analyzed following previously described methods with the following modifications (*Zhang et al., 2013*). Adaptor sequences were trimmed from reads using CutAdapt using default parameters. Reads were then further trimmed to remove the randomized adaptor dinucleotides on the 5′ and 3′ end. Read were next aligned to a database of mouse miRNA sequences (miRbase) using blastn with the following settings (blastn -word_size 11 -outfmt 6 -strand plus). Blast alignments were then parsed with custom python scripts to extract and count the best miRNA alignment for each read with a minimum read alignment of 18 nucleotides. miRNA counts for each library were then filtered to keep only miRNAs with a minimum count of 50 reads in two libraries and analyzed for differential expression using EdgeR, with classical analysis parameters.

## Affymetrix microarray analysis

For the *miR-203* epidermal loss-of-function microarray analysis, RNA was isolated from total epidermal samples from two-pairs of *miR-203*[+/+] and *miR-203*[−/−] animals at p4. For the doxycycline-inducible *miR-203* over-expression microarray analysis, RNA was isolated from two pairs of doxycline induced or uninduced *Krt14-rtTA/ pTre-miR-203* animals using FACS sorting for Krt14-H2B-GFP+ cells as described previously (*Jackson et al., 2013*). The microarray analysis of *miR-203* overexpression in basal epidermis was previously published (*Jackson et al., 2013*). Subsequently total RNAs (500 ng) were processed and hybridized to the GeneChip Mouse Genome 430 2.0 array (Affymetrix, Santa Clara, CA) following the manufacturer's instruction at the MCDB microarray facility. Microarray image files were processed using the R Bioconductor suite and Mas5 normalization. Probesets were then filtered to include only those probes with present or absent calls in at least two arrays. Probesets were then collapsed using the probeset with the maximum averaged probeset intensity to represent each GeneID. Log2 fold changes were then computed using the *limma* Bioconductor package.

## Ribosome profiling

Ribosome profiling was performed on primary keratinocyte lysates using the ARTseq Ribosome profiling kit (Illumina). Briefly, lysates from a 10-cm dish of primary keratinocytes were isolated in the presence of cycloheximide (Sigma-Aldrich, St. Louis, MI, 50 μg/ml) and subject to limited RNAse digestion (10 units) for 45 min at room temperature. RNase digestion was terminated by addition of 15 μl of SUPERase In (Thermo Fisher Scientific) followed by ribosome isolation using illustra MicroSpin S-400 HR Columns (GE Healthcare, United Kingdom). Following RNA extraction and precipitation, rRNA was depleted using the Ribo-Zero Gold kit (Illumina), with the remaining RNA then fractionated through 18% PAGE gels. RNA species 28–32 nt were isolated for adaptor ligation, reverse transcription, circularization, and PCR amplification following the manufacturer's protocol. PAGE gel isolated PCR products were then sequenced on a HiSeq 2000 (Illumina).

Raw reads were first trimmed to remove 3′ adaptors using cutAdapt with default parameters. Reads were aligned to mm10 rRNA, tRNA, and ncRNA (Ensemble annotation) databases using Bowtie (default settings) to exclude reads aligning to abundant rRNA, tRNA, and ncRNA sequences. Unaligned reads were then aligned via Tophat using default settings, with a supplied .gtf annotation file containing combined Refseq and Ensembl gene annotations (iGenomes Illumina downloaded 9/4/2013). Uniquely aligned read counts were quantified across each CDS using HTSeq Count (settings: -s yes -m union -t CDS) using the above-mentioned GTF annotation database. Transcripts with low reads counts were excluded by only keeping transcripts with at least 50 reads in at least two libraries. Filtered transcript reads count data were then analyzed for differential expression using EdgeR with classical analysis parameters. To calculate translation efficiency for each transcript, Reads Per Million Mapped (RPM) values from 3Seq were divided by Reads per Million Mapped to coding sequence for the Ribo-Seq. The change in translation efficiency was then computed as the ratio of translation efficiency in the *miR-203*[−/−] and *miR-203*[+/+] libraries.

## Ago2 HITS-CLIP

*Ago2* HITS-CLIP was performed as described with minor modifications (*Chi et al., 2009*). 15-cm$^2$ dishes of primary keratinocytes were irradiated twice at 200 mJ/cm with 254 nm UVC light. Following irradiation, cell lysates were harvested by scraping and stored at −80°C. Following thawing, lysates were further lysed by trituration 3 times through pre-chilled 25- and 30-gauge needles. Lysates were then treated with 10 µl Turbo DNase (Thermo Fisher Scientific) and 5 µl RNaseOUT (Thermo Fisher Scientific) per ml of lysate. Limited RNase digestion was performed using 10 µl per ml of lysate of a 1:20 dilution of an RnaseA/T1 mix (Sigma-Aldrich/Thermo Fischer Scientific 1× mix = 3.33 µl RnaseA [2 µg/µl] with 6.66 µl RnaseT1 [1 U/µl]). Crosslinked *Ago2* was recovered via immunoprecipitation for 2 hr at 4°C with 3 µg of a monoclonal anti-mouse *Ago2* antibody (Wako Chemicals USA Inc., Richmond, VA, clone 2D4) complexed with Protein-G Dynabeads (Thermo Fisher Scientific). Immunoprecipitates were washed twice with High-Salt Buffer and PNK buffer, then end labeled with 25 µCi $^{32}$P y-ATP using PNK 3′ phosphatase minus (New England Biolabs) for 5 min at 37°C. After washing the beads as listed above, 5′ adaptor ligation was performed for 2 hr at room temperate using T4 RNA Ligase 1 with 10 µM 5′ RNA Linker, 20% PEG-8000 (wt/vol final), 1 mM ATP, and RNaseOUT (Thermo Fisher Scientific). Beads were again washed twice with PNK buffer then resuspended in a phosphatase reaction with 5 µl FastAP (Thermo Fisher Scientific) with RNaseOUT. Following washing twice with PNK buffer, Protein–RNA complexes were eluted from the beads using 1× NuPage Loading buffer supplemented with 50 mM DTT at 70°C for 10 min. Protein–RNA complexes were then resolved on an 8% Bis-Tris Gel and transferred to nitrocellulose. Membranes were exposed to a phosphor screen for 1–2 hr to obtain an autoradiograph. Subsequently, RNA–protein complexes migrating in the 110–130 kD range were excised. RNA was recovered from the nitrocellulose using Proteinase K treatment followed by phenol–chloroform extraction and ethanol precipitation. Isolated RNA was ligated to a 3′linker using the same ligation reaction conditions as for 5′ ligation, and ligated RNA species were fractionated away from adaptor–adaptor products on 10% UREA PAGE gels. The RNA was eluted from the PAGE gel with HSCB buffer overnight at 4°C, then ethanol precipitated and resuspended for reverse transcription with SuperScript III (Thermo Fisher Scientific). cDNA products were then subjected to PCR amplification for 20–24 cycles and fractionated on an 8% native PAGE gel. PCR products representing cDNA inserts of 20–50 nts were recovered and subject to sequencing on a HiSeq 2000 (Illumina).

HITS-CLIP reads were analyzed as follows. First, reads were processed to identify miRNA alignments using the same pipeline that we use for Small-RNA-Seq listed above. Reads not mapping to miRNAs were next processed as follows. Reads were trimmed with Cutadapt to remove adaptor sequences using default settings. To avoid PCR duplicates from biasing the analysis, duplicate reads were then collapsed to a single read using Fastx_collapser (default settings). The 5′ and 3′ adaptor sequences contain randomized dinucleotides on their 3′ and 5′ ends respectively, which were next trimmed from the reads. The reads were then aligned to the mm10 genome assembly using NovoAlign requiring a minimum alignment length of 25 nucleotides (settings –s 1 –t 85 –l 25) (Novocraft, Malaysia). All unique alignments from each library were then pooled to identify *Ago2* HITS-CLIP clusters. Clusters were defined as two read alignments that overlap by a minimum of one nucleotide. Clusters were next annotated to gene features in a hierarchical manner in which clusters were annotated to protein coding RefSeq 3′ UTRs (with 5 kbp extension allowed), RefSeq CDS regions, RefSeq 5′ UTRs, Ensemble ncRNA regions, and RefSeq intron regions. Clusters not found in these regions were annotated as intergenic clusters. For predicting miRNA target sites, 3′UTR cluster sequences were searched for 6mer seed-sequence matches for miRNA species that accounted for 90% of miRNAs expressed in epidermis based on small-RNA sequencing. From this data set, 117 binding sites in 113 mRNAs were predicted be *miR-203* target sites.

## Meta-analysis of miR-203 targets

The *miR-203* overexpression data sets used in the meta-analysis included previously published microarrays from sorted H2B-GFP$^+$ epidermal cells with transient *miR-203* induction (GSE45121), and microarrays from sorted H2B-GFP$^{hi+}$ hair follicle progenitor cells with transient *miR-203* induction described in this manuscript. The *miR-203* knockout data sets used in the meta-analysis included the microarrays from p4.5 total epidermal samples from miR-203$^{+/+}$ and miR-203$^{−/−}$ animals, ribosome profiling of *miR-203*$^{+/+}$ and *miR-203*$^{−/−}$ primary keratinocytes, and 3seq of *miR-203*$^{+/+}$ and *miR-203*$^{−/−}$ primary cultures. We also performed 3seq on *miR-203*$^{−/−}$ cells transformed with *Hras*$^{G12V}$; however, there was no enrichment for *miR-203* seed matches in the upregulated transcripts based on CDF

analysis, consistent with the low levels of *miR-203* in *Hras*^G12V-transformed keratinocytes, and therefore, this data set was excluded from the expression meta-analysis (data not shown). Gene symbols were used to compare across microarray, 3Seq, and Ribo-Seq data sets. Log$_2$ fold changes were used to assess differential gene expression in each data set. In total 6407 genes were detectable in all the data sets, from which 294 satisfied the criteria of being upregulated in all the *miR-203* knockout data sets and downregulated in the *miR-203* overexpression data sets. Negative control data sets were constructed to analyze the enrichment of genes containing *miR-203* seed matches with the following criteria, randomly selected genes from the list of detected transcripts in the meta-analysis, or genes that have a positive correlation to *miR-203* expression (downregulated in *miR-203* knockout data sets and upregulated in *miR-203* overexpression data sets).

In order to compare gain-of-function and loss-of-function data sets in aggregate, a ranked correlation to *miR-203* metric was calculated. Transcripts were assigned a rank in each knockout data set with the most upregulated gene given a rank of 1. Transcripts were next assigned a rank in each overexpression data set with the most downregulated gene given a rank of 1. The ranked values from each of the 5 data sets were then summed and ranked with the transcript most upregulated upon *miR-203* ablation and downregulated upon overexpression being assigned a value of 1.

## Hierarchical clustering, GO-term analysis, GSEA, 3′UTR motif-searching

Normalized transcript abundances (expressed as Reads Per Million) were used for mean-centered unsupervised hierarchical clustering using Cluster 3.0 software. For 3seq data, only transcripts with at least twofold change were selected and for miRNA data, only miRNAs with at least an expression level of 1000 RPM and at least a twofold change were selected for visualization in JavaTree View. GO-term enrichment analysis was performed by DAVID using Gene-IDs as input, with analysis being performed using GO biological processes data sets. GSEA analysis was performed using ranked expression values for the displayed data set, and genesets were selected from the referenced publications. De novo motif searching was performed using the HOMER package to search for 7 or 8mer motifs in RefSeq 3′UTR sequences for selected genesets. For genes with multiple 3′UTR isoforms, the longest 3′UTR was selected for motif searching. miRNA seed-sequence searches in *Ago2*-HITS-CLIP clusters were performed with a Python script using regular expressions.

## Analysis of The Cancer Genome Atlas (TCGA) data

Head and Neck SCC miRNA-Seq data were obtained from the TCGA https://tcga-data.nci.nih.gov/tcga/ (Download date 10/23/2014). Patient matched normal solid tissue and tumor miRNA quantification records were identified using custom R scripts (regular expression query for tumor and solid tissue normal samples respectively 'TCGA-[0-9A-Z]{2}-[0-9A-Z]{4}-0', 'TCGA-[0-9A-Z]{2}-[0-9A-Z]{4}-11'). The normalized reads_per_million quantification for *miR-203* and *miR-21* was then plotted to determine the relative expression in normal and tumor tissue samples.

## Mouse strains and generation of the conditional miR-203 knockout mouse

A gene targeting vector was constructed that contained 11 kbp homologous region surrounding the *miR-203* locus (*Figure 3—figure supplement 1*). LoxP were inserted flanking the *pre-miR-203* sequences, with a neomycin selection cassette flanked by Frt sites. The construct was electroporated into Cy2.4 ES cells (B6(Cg)-Tyr<c2J> genetic background). Positive clones were identified by Southern blot analysis using a probe complementary to the 3′ end of the targeted homologous region. ES cells were injected into blastocysts and chimeras were screened based on white/black coat color selection. Upon obtaining germline transmission, the neo cassette was excised by breeding the F1 progeny to an *Actb-Flpe* line maintained on a C57BL/6J background (obtained from Jackson Labs, Bar Harbor, ME). *miR-203*^floxed animals were then bred to a *EIIa:Cre* line maintained on a C57Bl/6 background (obtained from Jackson Laboratory) or a *Krt14:Cre* line maintained on a mixed background (obtained from E. Fuchs Laboratory), to obtain germline or conditional ablation of *miR-203*. The *EIIa:Cre* transgene was removed from the germline *miR-203* deleted line by backcrossing to a C57Bl/6 line and subsequently maintained on a C57Bl/6 background. *pTre2-miR-203/Krt14-rtTA* mice were generated as described previously and maintained on an FVB background (*Jackson et al., 2013*).

Mice were bred and housed according to the guidelines of IACUC at a pathogen-free facility at the University of Colorado (Boulder, CO, USA).

## Immunofluorescence, EdU detection, and miRNA in situ hybridization

Frozen cryostat sections (8 µM) were fixed in 4% paraformaldehyde for 10 min at room temperature, washed thrice with PBS, and blocked for 10 min using Gelatin Block (0.1% Triton X-100, 2% gelatin, 2.5% normal goat serum, 2.5% normal donkey serum, and 1% BSA in PBS). Primary antibodies, diluted in gelatin block, were then incubated overnight (see *Supplementary file 2* for antibody references). Following three washes with PBS, sections were incubated with appropriate Alexa-Fluor secondary antibodies (1:2000) for 1–2 hr at room temperature. Following three washes with PBS, sections were stained with Hoescht Dye and mounted in Anti-fade solution. miRNA in situ hybridization for *miR-203* was performed on frozen sections as described previously (*Yi et al., 2008*). EdU detection was performed following manufacturer's instructions, with the following parameters. P4 animals were IP injected with 50 µg/g EdU 4 hr prior to tissue harvest. Following EdU detection, the sections were blocked and probed with antibodies as described above. BrdU detection was performed as previously described, with the following parameters. Pregnant female mice were IP injected with 50 µg/g BrdU for 2 hr prior to embryo harvest in OCT compound. *miR-203* in situ hybridization on FFPE mouse and human tumor samples was performed with the following modifications, after deparaffinization in xylenes, the tissue was treated with proteinase K (20 µg/ml) for an extended period of 20 min at an elevated temperature (37˚C). Microscopy images were obtained using a Leica DM5500B microscope with either a Leica camera (brightfield) or Hamamatsu C10600-10B camera (fluorescence) and processed with the Leica image analysis suite, MetaMorph (MDS Analytical Technologies, Sunnyvale, CA) and Fiji software. BrdU or EdU image quantifications were performed by counting the number of Krt5+/EdU or BrdU positive cells in randomly chosen microscopy fields. The length of the basement membrane was used to represent the length of the epidermis analyzed and was determined by tracing the basement membrane and calculating line length using Fiji software. Epidermal thickness was assessed by tracing a line tangential to the basement membrane and extending to the beginning of the stratum corneum and calculating the line length.

## qPCR and western blotting

qPCR was performed using the Qiagen (Germany) miR-script RT system and a BioRad CFX-384 machine (Hercules, CA). Fold-changes were computed using the ΔΔCt formula normalized to *sno25* and *Hprt* values. In all qPCR figures, error bars denote standard errors of the normalized mean. Western blotting was performed using 20–40 µg of protein lysate run on 8–12% SDS-PAGE gels. Proteins were transferred to PVDF for detection of *Pola1*, *β-tubulin (Tubb5)*, or *Ccnd1*. Primary antibodies were incubated in 5% BSA overnight and detected using HRP-conjugated secondary antibodies and Amersam ECL-Plus reagents (1:10,000)(GE Healthcare). See *Supplementary file 2* for antibody descriptions and dilutions. X-ray films were scanned and processed with Fiji software to calculate relative protein abundance.

## Primary keratinocyte harvesting and cell culture, viral infections, Edu flow cytometry, and shRNA knockdown

Primary keratinocytes were isolated from neonatal mice using previously described methods with the following modifications (*Lichti et al., 2008*). Isolated skin was incubated on a solution of Dispase overnight at 4˚C to dissociate the epidermis from the dermis. The following day epidermal sheets were incubated in 37˚C Trypsin for 10 min to isolate keratinocytes. Primary keratinocytes were then plated in 6-cm or 10-cm dishes with E-Low media supplemented with 0.2 mM calcium chloride for the first 24 hr then switched to E-Low media with 0.05 mM Ca++. Lentiviral particles were produced by transient transfection of pLKO-shRNA constructs, PsPax.2, and pVSVG. 24 hr post transfection, the media was changed to E-Low calcium. Retroviral particles were produced by transient transfection of pBabe-vector-puro, pBabe-Hras^G12V-puro, pBabe-vector-neo, or pBabe-Hras^G12V-neo, with PCL-Eco and pAdvantage packaging plasmids. Viral supernatant was harvested every 12 hr for up to 96 hr, pooled and filtered with 0.45-µM filter. *Ad-eGFP* or *Ad-CREeGFP* adenoviruses were obtained from the Iowa Gene Transfer Core and used at MOI of 50. Retroviral and lentiviral infections were performed 3–4 days after plating primary keratinocytes. Keratinocytes were selected with 2 µg/ml puromycin for 48 hr or 50 µg/ml neomycin for 7 days, at which time non-infected cell cultures were

non-viable. Spontaneously immortalized *miR-203*$^{+/+}$, *miR-203*$^{-/-}$, and *miR-203*$^{fl/fl}$ mouse keratinocyte cell lines were also generated via serial passage on mitomycin-C treated NIH-3T3 feeder cell culture layer and utilized for assays as noted in the text. Flow cytometry was performed as previously described, with minor modifications (*Jackson et al., 2013*). Cell cultures derived from *pTre2-miR-203/Krt14-rtTA* animals were treated with 5 μg/ml doxycycline for 24 hr to induce *miR-203* expression, pulsed with EdU (10 μM) for 30 min, harvested and analyzed according to the Click-IT EdU Plus instructions on an BD Cyan flow cytometer (Thermo Fisher Scientific). For colony formation assays, 2000 cells were split into individual wells of 6-well plates, cultured for 10–14 days, fixed with 4% PFA, and stained with 0.2% crystal violet in 70% ethanol. For induction of *miR-203*, 24 hr after plating, cells were supplied with fresh media containing doxycycline at 5 μg/ml. Sigma–Aldrich TRC lentiviral shRNAs against *Hbegf* and *Pola1* were obtained from the Functional Genomic Facility (University of Colorado at Boulder, sequences listed in *Supplementary file 2*).

## 3′UTR luciferase assays

3′UTR reporter constructs were generated by PCR amplification of 3′UTRs from cDNA or gDNA and subcloning of the fragments into pGL3-Control (Promega, Madison, WI) (primer sequences listed in *Supplementary file 2*). 2 ng renilla luciferase control, 20 ng pGL3-3′UTR reporter, and 380 ng of Krt14 empty vector, or Krt14-*miR-203* were transiently cotransfected into keratinocytes in each well of a 12-well plate using Mirus Bio LT1 reagent (Mirus Bio LLC, Madison, WI). 24 hr later cell lysates were collected, and renilla and firefly luciferase activity were measured using Dual-Glo Luciferase Assay system (Promega) as described previously (*Yi et al., 2008*). Data are represented as the ratio of firefly to renilla RFU values, normalized to Pgl3-control values. Error bars represent propagated standard deviations.

## DMBA/TPA carcinogenesis

DMBA/TPA carcinogenesis was performed as described previously (*Abel et al., 2009*). The backskin of 7- to 9-week-old *miR-203*$^{+/+}$ and *miR-203*$^{-/-}$ mice was shaved. 48 hr later the backskin was painted with a single dose of 25 μg of DMBA in 200 μl of Acetone. 2 weeks following DMBA treatment the mice began receiving bi-weekly treatments of 4 μg of TPA in acetone. The number of palpable tumors of at least 1 mm in diameter, persisting for at least 2 weeks, was recorded weekly. Tumor diameters were measured using a digital caliper. Following 21 weeks of TPA treatment, mice were euthanized and tumors were collected for *Hras*$^{Q61I}$ genotyping, OCT embedding, and paraffin embedding.

## Hras$^{Q61L}$ genotyping of DMBA/TPA tumors

Tumor DNA was isolated by incubating the tissue in a DNA Lysis Buffer (400 mM NaCl, 0.1% SDS, 1 mM EDTA, 1 μg/ml Proteinase K) at 55°C for 4 hr. Lysates were then vortexed and lightly centrifuged to liberate DNA from the partially digest tumor tissue. The supernatant was then removed and subject to Phenol–chloroform extraction, followed by isopropanol precipitation. Isolated DNA pellets were then resuspended in TE buffer and quantified by UV spectrophotometry (10 mM Tris pH 8.0, 1 mM EDTA). The *Hras* gene was PCR amplified using primers that flank exon 2. Following amplification, the PCR reactions were digested with 5 units of XbaI restriction enzyme at 37°C. The reaction products were then resolved and visualized on a 3% agarose gel. DNA isolated from the tails of animals in the DMBA/TPA experiment was treated in parallel as a negative control for detection of the *Hras*$^{Q61L}$ mutation.

## Statistical analysis

Statistical analysis was performed using either R or Microsoft Excel. Statistical methods employed are indicated in the figure legends. Unpaired two-sided Student's *t*-tests were used to assess statistical significance unless indicated otherwise in the figure legends. For comparisons with multiple categories, ANOVA was used with Tukey's HSD post-hoc test. Non-parametric Whitney–Mann U-tests were used to assess significance for the tumor multiplicity measurements (*Abel et al., 2009*). The hypergeometric test was used to assess the enrichment of gene lists in genome-wide studies. The Kolmogorov–Smirnov test was used to assess differences in cumulative distributions functions.

## Data access

All sequencing and microarray data are deposited in the Gene Expression Omnibus (GSE66056).

## Acknowledgements

We thank C Yang, J Gao, and D Feng for the generation of the *miR-203* KO mouse. We thank members of the Yi laboratory for their critical discussions. We also thank P Muhlrad for critical reading of the manuscript.

## Additional information

### Funding

| Funder | Grant reference | Author |
|---|---|---|
| National Institute of Arthritis and Musculoskeletal and Skin Diseases (NIAMS) | R01AR059697 | Kent Riemondy, Rui Yi |
| American Cancer Society (ACS) | 124718-RSG-13-197-01-DDC | Rui Yi |
| National Institute of Arthritis and Musculoskeletal and Skin Diseases (NIAMS) | R01AR066703 | Rui Yi |
| National Cancer Institute (NCI) | R01CA052607 | Enrique C Torchia, Dennis R Roop |
| National Institute of Arthritis and Musculoskeletal and Skin Diseases (NIAMS) | P30AR057212 | Kent Riemondy, Dennis R Roop, Rui Yi |

The funders had no role in study design, data collection and interpretation, or the decision to submit the work for publication.

### Author contributions

KR, RY, Conception and design, Acquisition of data, Analysis and interpretation of data, Drafting or revising the article; X-W, DRR, Acquisition of data, Drafting or revising the article, Contributed unpublished essential data or reagents; ECT, Acquisition of data, Contributed unpublished essential data or reagents

### Ethics

Animal experimentation: This study was performed in strict accordance with the recommendations in the Guide for the Care and Use of Laboratory Animals of the National Institutes of Health. All of the animals were handled according to approved institutional animal care and use committee (IACUC) protocols (#1408.01) of the University of Colorado, Boulder. Every effort was made to minimize suffering.

## Additional files

### Supplementary files

• Supplementary file 1. Sequencing mapping statistics.

• Supplementary file 2. Primers, antibodies, and shRNAs used in this study.

### Major dataset

The following dataset was generated:

| Author(s) | Year | Dataset title | Dataset ID and/or URL | Database, license, and accessibility information |
|---|---|---|---|---|
| Riemondy K, Yi R | 2015 | MicroRNA-203 represses selection and expansion of oncogenic HRas transformed tumor initiating cells | http://www.ncbi.nlm.nih.gov/geo/query/acc.cgi?acc=GSE66056 | Publicly available at the NCBI Gene Expression Omnibus (Accession no: GSE66056). |

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
