## [Decision Letter]

Thank you for sending your work entitled “*MicroRNA-203* represses selection and expansion of oncogenic *HRas* transformed tumor initiating cells” for consideration at *eLife*. Your article has been favorably evaluated by Fiona Watt (Senior editor) and two reviewers, one of whom, Chi Dang, is a member of our Board of Reviewing Editors.

The Reviewing editor has compiled the following critique and list of specific comments to help you prepare a revised submission:

The manuscript by Riemondy et al. describes an analysis of mRNA and miRNA expression in mouse keratinocytes transformed by *Hras*^*G12V*^ expression. They report the repression of *miR-203* by oncogenic H-Ras in squamous cell carcinoma of the skin, purporting that the stem cell pool is affected. This work emanates from previous work connecting *miR-203* to skin stem cell biology. Through loss and gain-of-function studies, the authors document the tumor suppressive role of *miR-203*, however, the mechanistic basis for the growth inhibitory role of *miR-203* is less well established in this study. The authors performed a large number of transcriptome-wide analyses on *miR-203*-deficient and overexpressing cells (RNA-seq, microarray, ribosome profiling, CLIP-seq) in an attempt to identify the mechanisms underlying the phenotypic effects they observed. Unfortunately, these studies were not highly informative, especially in the case of *miR-203*-/- cells, where only very minor gene expression changes were observed. Two targets of *miR-203* were documented and their loss of function diminishes cell viability, but whether this is H-Ras specific is not established. That is, whether knockdown of *Pola1* affect non-transformed cells was not specifically addressed, since the authors suggest that *Pola1* is involved in the 'Ras signaling pathway.' The mechanism by which activated H-Ras silence *miR-203* expression was also not addressed. Notwithstanding these concerns, the work appears quite extensive and thorough and therefore could be suitable for publication in *eLife*. The strength of the paper is the novel demonstration, through genetic deletion studies, that *miR-203* functions as a bona fide tumor suppressor in vivo. These findings are nicely correlated with in vitro data, establishing the ability of *miR-203* to modulate *Hras*-mediated transformation.

1) The authors should simplify their presentation of the genomic data and be more strategic about which datasets they include in a revised manuscript, including only those datasets and bioinformatic analyses that offer clear mechanistic insight.

2) More mice should be analyzed in Figure 3, especially at the E16 time-point when only 2 embryos were examined. The trend towards increased BrdU/EdU incorporation may then become significant, which would be an important finding since this is the only potential phenotype in the intact knockout animals. With the currently presented data, it is an overstatement to claim that “*miR-203* functions as a gatekeeper to limit cell division when the proliferation rate is high during early embryonic skin development…”

3) Where are the data that show that “genes involved in regulation of mitotic cell cycle, DNA synthesis, and metabolism were prominently downregulated upon *miR-203* induction” (the subsection “Comprehensive identification of *miR-203* targets reveals regulation of the Ras signaling pathway”, second paragraph)? Was an unbiased method such as GSEA or gene ontology employed?

4) Based on the data presented in this manuscript, it is an overstatement to claim that “*miR-203* directly targets the Ras-mediated signaling pathway” (end of the subsection “Comprehensive identification of *miR-203* targets reveals regulation of the Ras signaling pathway”). At best, a few targets that contribute to multiple pro-proliferative pathways were identified. It is plausible that *miR-203* exerts its effects via the coordinated, although minor, downregulation of a broad set of targets, some of which include members of the Ras pathway.

5) shRNA knockdown experiments with putative targets (Figure 7) should be performed in *Hras*-transformed *miR-203* +/+ and -/- cells to see if these targets are relevant to transformation in these settings.

---

## [Author Response]

1) The authors should simplify their presentation of the genomic data and be more strategic about which datasets they include in a revised manuscript, including only those datasets and bioinformatic analyses that offer clear mechanistic insight.

We thank you for the suggestion. We have now revised Figure 6, in which we used an extensive set of genomic analyses to interrogate the impact of *miR-203* on the transcriptome and identify high-confidence *miR-203* targets. Specifically, we moved Figure 6 to Figure 6—figure supplement 1, Figure 6—figure supplement 2, Figure 6—figure supplement 3 and Figure 6—figure supplement 4. These data were utilized to synthesize our conclusion. By relocating them to figure supplements, we simplified our main Figure 6 to present the results of our transcriptome analysis and HITS-CLIP analysis for identifying *miR-203* targets. In addition, we added another Figure 6—figure supplement 3 to document the strong overlap between *Ago2* footprint and miRNA seed matches and list all detected miRNA seed matches detected in our experiments. These data further validated our HITS-CLIP study and provided additional data for readers, who may be interested in miRNA targets in keratinocytes.

*2) More mice should be analyzed in*
Figure 3*, especially at the E16 time-point when only 2 embryos were examined. The trend towards increased BrdU/EdU incorporation may then become significant, which would be an important finding since this is the only potential phenotype in the intact knockout animals. With the currently presented data, it is an overstatement to claim that* “*miR-203 functions as a gatekeeper to limit cell division when the proliferation rate is high during early embryonic skin development…*” *(the subsection “Genetic deletion of miR-203 impacts early epidermal development in mouse”, second paragraph)*.

During our revision, we have set up more than 6 breeding pairs in an effort to produce *miR-203* KO embryos. In total, we have generated one pair of E16 WT and KO embryos, and one pair of E17 WT and KO embryos because of the small litter size and our heterozygous to heterozygous breeding strategy, which has 25% chance to produce KO. Nevertheless, we incorporated these new embryonic data to our existing data in Figure 3. We now have three pairs of E16 embryos and four pairs of E17 pairs. The p-value for the E16, BrdU/EdU incorporation study is 0.07 and the p-value for the E17, BrdU/EdU incorporation study is 0.13. In addition, the p-value for the E16 epidermal thickness study is 0.03 and the p-value for the E17, epidermal thickness study is 0.01. The new data are consistent with our original observations, that is *miR-203* KO epidermis are more proliferative than the WT counterparts. Together, these data further strengthened our original observations.

We also revised our statement to “*miR-203* limits cell division when the proliferation rate is high during early embryonic skin development”, to more accurately reflect the observed defects.

*3) Where are the data that show that* “*genes involved in regulation of mitotic cell cycle, DNA synthesis, and metabolism were prominently downregulated upon miR-203 induction*” *(the subsection “Comprehensive identification of miR-203 targets reveals regulation of the Ras signaling pathway”, second paragraph)? Was an unbiased method such as GSEA or gene ontology employed?*

These statements were based on unbiased gene ontology analyses, which were presented in the panel of “*miR-203* inducible analysis” in “[Supplementary-material SD2-data]_GO analysis.” However, we didn’t link this source in our original manuscript and we apologize for the negligence. In the revised manuscript (the subsection “Comprehensive identification of *miR-203* targets reveals regulation of the Ras signaling pathway”, third paragraph), we stated “GO-analysis demonstrated that genes involved in regulation of mitotic cell cycle, DNA synthesis and metabolism were prominently downregulated upon *miR-203* induction ([Supplementary-material SD2-data]).”

*4) Based on the data presented in this manuscript, it is an overstatement to claim that* “*miR-203 directly targets the Ras-mediated signaling pathway*” *(end of the subsection “Comprehensive identification of miR-203 targets reveals regulation of the Ras signaling pathway”). At best, a few targets that contribute to multiple pro-proliferative pathways were identified. It is plausible that miR-203 exerts its effects via the coordinated, although minor, downregulation of a broad set of targets, some of which include members of the Ras pathway*.

We agree with the reviewers’ assessment. Based on our data and the reviewers’ suggestion, we revised our conclusion (in the subsection “Comprehensive identification of *miR-203* targets reveals regulation of the Ras signaling pathway”) to “This collection of *miR-203* targets suggests that *miR-203* targets multiple pathways, including several components of the Ras signaling pathway, to suppress cell proliferation.”

*5) shRNA knockdown experiments with putative targets (*Figure 7*) should be performed in Hras-transformed miR-203 +/+ and -/- cells to see if these targets are relevant to transformation in these settings*.

We carried out the shRNA knockdown experiments for both *Pola1* and *Hbegf* in *HRas*-transformed miR-203 WT and KO keratinocytes as suggested by the reviewers. In both cases, we observed strong reduction in colony forming capacity of these transformed cells, similar to our original observations with non-transformed keratinocytes. Collectively, these data further support the requirement of *Pola1* and *Hbegf* in cell proliferation/survival of both *HRas* transformed and non-transformed cells. We present these data as Figure 7—figure supplement 2 to in the revised manuscript.